# Post-transcriptional reprogramming by thousands of mRNA untranslated regions in trypanosomes

Anna Trenaman[1,3], Michele Tinti [1,3], Richard J. Wall[1,2] & David Horn [1] ✉

Although genome-wide polycistronic transcription places major emphasis on post-transcriptional controls in trypanosomatids, messenger RNA *cis*-regulatory untranslated regions (UTRs) have remained largely uncharacterised. Here, we describe a genome-scale massive parallel reporter assay coupled with 3′-UTR-seq profiling in the African trypanosome and identify thousands of regulatory UTRs. Increased translation efficiency was associated with dosage of adenine-rich poly-purine tracts (pPuTs). An independent assessment of native UTRs using machine learning based predictions confirmed the robust correspondence between pPuTs and positive control, as did an assessment of synthetic UTRs. Those 3′-UTRs associated with upregulated expression in bloodstream-stage cells were also enriched in uracil-rich poly-pyrimidine tracts, suggesting a mechanism for developmental activation through pPuT 'unmasking'. Thus, we describe a *cis*-regulatory UTR sequence 'code' that underpins gene expression control in the context of a constitutively transcribed genome. We conclude that thousands of UTRs post-transcriptionally reprogram gene expression profiles in trypanosomes.

Post-transcriptional regulatory sequences often reside within messenger RNA (mRNA) 3′-untranslated regions (UTRs). UTRs are thought to be particularly important in the context of unregulated polycistronic transcription in the trypanosomatids, where differential expression controls necessarily operate almost exclusively post-transcription[1], at the level of mRNA abundance and translation control in particular[2–6]. Although functional characterisation of trypanosomatid protein-coding regions has progressed rapidly in recent years, there has been less progress in the characterisation of mRNA untranslated regions which, based on our updated annotation, comprise approx. 34% of the African trypanosome genome.

The parasitic trypanosomatids include the African and South American trypanosomes and the *Leishmania* spp[7]. African trypanosomes, *Trypanosoma brucei*, cause sleeping sickness, or human African trypanosomiasis; American trypanosomes, *Trypanosoma cruzi*, cause Chagas disease; and the *Leishmania* spp. cause the leishmaniases.

Beyond these neglected tropical diseases, African trypanosomes also cause nagana in cattle and other livestock. These protozoan parasites are transmitted among mammals by distinct insect vectors. Gene expression control mechanisms in trypanosomatids are of interest, therefore, in terms of understanding pathogenesis, and in terms of developing interventions. The trypanosomatids also present notably tractable models for studies focussing on post-transcriptional controls operating in eukaryotes.

In trypanosomatids, long polycistronic transcription units contain genes encoding proteins with unrelated functions and with wide-ranging expression levels. Primary transcripts are co-transcriptionally processed into individual mRNAs by *trans*-splicing of a common, 39-nucleotide, capped 'spliced leader', and by polyadenylation[8–10]. These mRNA processing steps are coupled, and driven by 8–25 nucleotide, uracil-rich, poly-pyrimidine tracts, that are typically 80–140 nucleotides downstream of a polyadenylation site and 10–40

[1]The Wellcome Centre for Anti-Infectives Research, School of Life Sciences, University of Dundee, Dow Street, Dundee DD1 5EH, UK. [2]Present address: London School of Hygiene & Tropical Medicine, Keppel Street, London WC1E 7HT, UK. [3]These authors contributed equally: Anna Trenaman, Michele Tinti. ✉e-mail: d.horn@dundee.ac.uk

nucleotides upstream of a *trans*-splicing site[11]. No further consensus sequences have been defined for polyadenylation sites, typically an A; splicing branch sites, typically an A; or *trans*-splicing acceptor sites, typically the first AG dinucleotide downstream of the polypyrimidine tract.

The majority of experimentally characterised trypanosomatid UTRs are from *T. brucei*, and studies have typically focussed on 3′-UTRs that impact developmentally regulated expression; primarily for those proteins that substantially differ in abundance between the mammalian bloodstream-form of the parasite and the procyclic form, typically found in the mid-gut of the insect vector. Among approximately twenty-five genes with known regulatory 3′-UTRs are those encoding enzymes involved in ATP production[12–14], cell surface transporters[13,15–18], and mRNA-binding proteins[12,19,20]. Attention has also focussed on regulatory 3′-UTRs that impact the expression of other developmental stage-specific cell surface proteins[21], including genes that are unusually transcribed by RNA polymerase I, and that encode super-abundant surface proteins[22] and variant surface glycoproteins[23]. Despite intense efforts, however, regulatory 3′-UTRs, and the relevant *cis*-regulatory sequences, remain largely uncharacterised in the trypanosomatids.

To gain further insight into the principles governing post-transcriptional gene expression controls operating in trypanosomatids, we designed a Massive Parallel Reporter Assay (MPRA) in *T. brucei*. Implementation of a UTR-seq strategy allowed us to assess 3′-UTR regulatory potential at a genomic scale and revealed thousands of regulatory 3′-UTRs. Analysis of thousands of hit-fragments, and an independent assessment of native UTRs using a machine learning approach, revealed poly-purine tracts associated with increased translation. Our findings further suggest developmental modulation of poly-purine tract function by poly-pyrimidine tracts.

## Results

### A Massive Parallel 3′-UTR Reporter Assay

To develop a genome-scale screening platform for the identification of regulatory 3′-UTRs in *T. brucei*, we assembled the pRPai^UTR reporter construct, containing a dual positive (blasticidin deaminase, BSD) and negative (thymidine kinase, TK) selectable marker (Fig. 1a). The *BSD-TK* reporter was assessed using known positive or negative regulatory 3′-UTR fragments inserted immediately downstream; from aldolase[13] or cytochrome oxidase genes[14], respectively (Fig. 1a). The resulting constructs were used to derive recombinant *T. brucei* strains, which were assessed using drug-selection and dose-response analysis, revealing relative blasticidin-resistance and ganciclovir-sensitivity associated with use of the aldolase 3′-UTR, as intended (Fig. 1b). Inducible reporter expression, and increased expression driven by the aldolase 3′-UTR, was also confirmed by protein blotting (Supplementary Fig. 1a). The cassette immediately downstream of the *BSD-TK* stop-codon in pRPai^UTR then facilitated high-efficiency ligation of a library of *T. brucei* genomic DNA fragments, and assessment of both library complexity and fragment orientation (Fig. 1c). We constructed the library by cloning *T. brucei* genomic DNA fragments of 1–3 kbp in length and found that >90% of the clones analysed contained inserts in the expected size-range (Supplementary Fig. 1b); the semi-filling approach used maximised ligation of individual *T. brucei* genomic DNA fragments to the pRPai^UTR reporter construct (Fig. 1c). The plasmid library was then used to generate a high-complexity bloodstream form *T. brucei* library comprising approx. 2.5 million clones; given a haploid genome size of ~40 Mbp, this is equivalent to a theoretical >50 times genome coverage for both native oriented and inverted fragments (Supplementary Fig. 1c).

Library screening was carried out by pre-inducing reporter expression with tetracycline for 24 h, followed by either positive or negative selection (Fig. 1d). In the blasticidin, 'positive-control' arm of the screen, survival was increased by 3′-UTR fragments that increased BSD-TK expression; this is because blasticidin deaminase inactivates the blasticidin toxin. In contrast, in the ganciclovir, 'negative-control' arm of the screen, survival was increased by 3′-UTR fragments that reduced BSD-TK expression; this is because thymidine kinase converts ganciclovir into a toxin. We monitored growth during selection and harvested cells after four, six and eight days in each case; comparison with an unselected culture revealed substantial selection by both blasticidin, which reduced relative cell number >4000-fold by day eight, and ganciclovir, which reduced relative cell number >60-fold by day eight (Supplementary Fig. 1d). Genomic DNA was extracted from each selected sample, and DNA libraries were generated by amplifying the fragments cloned immediately downstream of the reporter, including the flanking index sequences (Fig. 1c, Supplementary Fig. 1e). A control set of amplicons was also generated in parallel using the plasmid library as template. All seven samples were deep-sequenced, and 'UTR-seq' reads were mapped to the *T. brucei* genome; 100.3 million paired reads for the blasticidin-selected samples (4.8% with an index sequence), 176.2 million for the ganciclovir-selected samples (4.8% with an index sequence), and 50.7 million for the plasmid library (6.5% with an index sequence). As a preliminary quality control step, we evaluated the read counts in annotated 3′-UTR regions and visualized the results using principal component analysis. This analysis revealed a cluster of blasticidin-selected samples and a separate cluster of ganciclovir-selected samples, which were both separated from the plasmid library control sample (Fig. 1e). We also quantified indexed read-counts adjacent to Sau3AI sites in the *T. brucei* genome. This analysis yielded signals for >48,000 sites, with strong enrichment adjacent to these sites, as expected given our use of Sau3AI to construct the library, and a substantial change in the profile following blasticidin-selection (Fig. 1f). These results are consistent with the view that distinct genomic fragments that impact reporter expression were enriched in the MPRA.

To visualise genome-wide hit-profiles, we identified enriched regions (peaks) for which the read count was increased following selection relative to the plasmid library control sample. We considered protein coding sequence (CDS)-derived peaks and peaks that extended outside of CDSs separately, and only reads associated with fragments cloned in their native orientation relative to the reporter at this stage; using indexed reads to determine orientation and relative enrichment. Positive selection with blasticidin yielded 80 enriched CDS peaks and 1827 enriched 'inter-CDS' peaks, while negative selection with ganciclovir yielded 918 enriched CDS peaks and 1915 enriched 'inter-CDS' peaks (Fig. 2a). The striking excess of 'inter-CDS' peaks relative to CDS peaks following positive selection with blasticidin suggested that fragments incorporating 3′-UTRs selectively supported robust reporter expression. To further explore this hypothesis, we visualised read-mapping relative to annotated genes and show three examples of long CDSs, encoding dynein heavy chains, that are on average twenty-five times longer than their cognate 3′-UTRs; these examples serve to illustrate the striking enrichment for positive regulatory fragments that do indeed incorporate 3′-UTRs (Fig. 2b).

A closer inspection of enriched inter-CDS peaks revealed some additional notable features, including association of subtelomeric Variant Surface Glycoprotein (*VSG*) genes with an increased proportion of positive regulatory fragments (Supplementary Fig. 2), as expected, given the positive regulatory potential of the *VSG* 3′-UTR[23]. In contrast, we observed an increased proportion of fragments linked to negative regulation derived from subtelomeric retrotransposon hotspot (*RHS*) arrays, *VSG* expression site associated gene (*ESAG*) arrays, and from *rDNA* loci (Supplementary Fig. 2). These observations are consistent with previously reported negative control of *RHS* genes[24] and a lack of mRNA processing signals at *rDNA* loci. We conclude that distinct genomic fragments that impact reporter expression were indeed enriched in the MPRA and that many 3′-UTR sequences likely function similarly in their native context and in the MPRA.

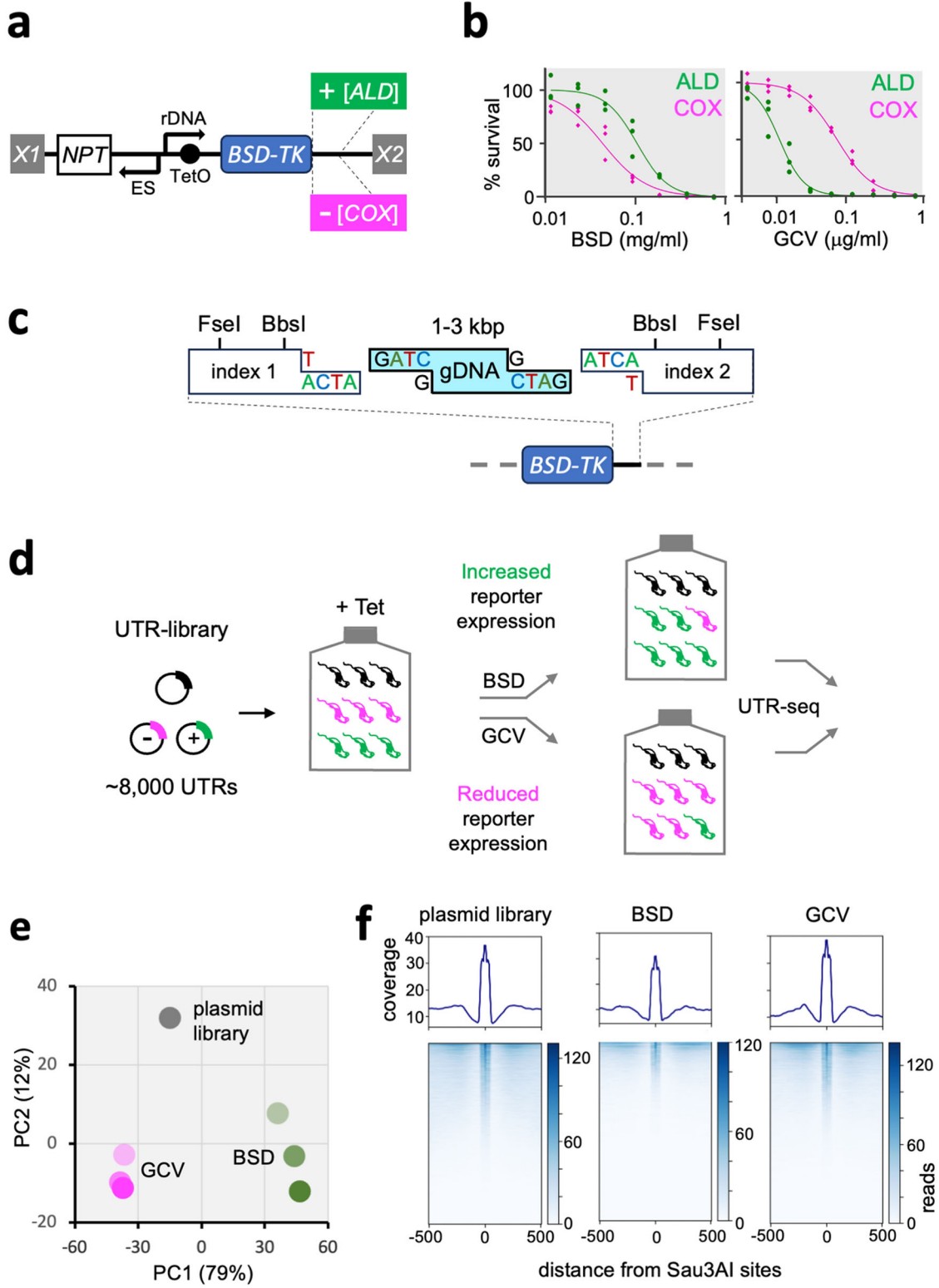

At this point, many *T. brucei* 3'-UTRs remained unannotated. Taking advantage of the Apollo annotation app[25] implemented at TriTrypDB[26], we revised or added 3'-UTR annotations for 4703 genes (see Methods for more details). Using the new data, now incorporating 3'-UTR annotations for >8028 genes, and >97% of all genes, we find the average 3'-UTR to be 622 b (median 388 b). Since the average CDS is 1576 b (median 1215 b), the average 5'-UTR is 177 b (median 82 b; excluding the common 39 b spliced leader sequence), and only 1974 b is thought to be dedicated to intronic sequence in just two CDSs in the

entire haploid genome, UTRs account for approx. 34% of the mRNA transcriptome in *T. brucei*, while 3'-UTRs account for approx. 26% (Fig. 2c).

### Identification of thousands of regulatory 3'-UTRs

All inter-CDS peaks recovered from the MPRA were manually curated, using indexed reads to define the boundaries of genomic fragments (see Methods for more details). This yielded 1941 and 2282 fragments that overlapped with 3'-UTRs, following positive selection with

**Fig. 1 | A massive parallel 3'-UTR reporter assay. a** The pRPa[iUTR] reporter construct. A blasticidin S-deaminase (*BSD*) and thymidine kinase (*TK*) fusion gene, with positive or negative regulatory 3'-UTRs cloned immediately downstream of the stop codon, was placed under the control of a tetracycline-inducible *rDNA* promoter and flanked by homology regions (*X1* is *HYG*[4] and *X2* is *rDNA*) to integrate the full construct at the tagged *rDNA* spacer locus in the 2T1 *T. brucei* strain; both RNA polymerase I and RNA polymerase II are used to drive protein coding gene transcription in *T. brucei*. A constitutively expressed *NPT* cassette under the control of a bloodstream *VSG* expression site (ES) promoter was also included to allow selection of recombinants in the absence of tetracycline. **b** Dose response curves reveal relative blasticidin (BSD) resistance and ganciclovir (GCV) sensitivity when reporter expression is increased (green); see Supplementary Fig. 1a. **c** The cassette immediately downstream of the *BSD-TK* stop-codon facilitated high-efficiency library construction. pRPa[iUTR] was digested with BbsI and T semi-filled, while *T. brucei* genomic DNA was partially digested with Sau3AI and fragments of 1–3 kbp were G semi-filled prior to ligation. The FseI sites and index sequences facilitated assessment of library complexity and fragment orientation. **d** The massive parallel reporter assay. The plasmid library was used to assemble a *T. brucei* library, which was induced with tetracycline, selected with BSD or GCV, and subjected to UTR-seq. **e** We sampled the library at days 4, 6 and 8 (lighter to darker shading), extracted genomic DNA, amplified cloned library fragments by PCR, deep-sequenced the products, mapped reads to annotated 3'-UTRs, and compared the outputs using principal component analysis. **f** Mapped reads adjacent to 48,509 Sau3AI sites in the *T. brucei* genome were quantified. Data for the plasmid library and sequences recovered following 6 days of selection are shown.

blasticidin, or negative selection with ganciclovir, respectively. All these recovered fragments were assigned to their cognate 3'-UTR and assessed for relative enrichment (Supplementary Data 1). We derived counts for indexed paired reads and ran pairwise comparisons for day-4, day-6 and day-8 positive and negative selected samples to calculate $\log_2$ fold changes. Given the absence of downstream mRNA processing signals in our reporter construct, we expected relatively few mRNA processing signals and regulatory elements generated by fragments cloned in the inverted orientation relative to the reporter. Indeed, a comparison of $\log_2$ fold changes for fragments cloned either in the native orientation, or in inverted orientation, revealed the expected strand-bias following either positive ($p = 2.4^{-113}$) or negative ($p = 2.7^{-21}$) selection (Fig. 3a).

To further explore the quality and coverage of our dataset, we asked whether previously described regulatory 3'-UTRs in *T. brucei* registered enriched hit fragments in the appropriate arm of the screen. A literature search yielded twenty-five documented regulatory 3'-UTRs; thirteen linked to positive regulation and twelve linked to negative regulation in bloodstream form trypanosomes (Supplementary Data 1). Nine of these known regulatory 3'-UTRs register as hits in the expected positive arm of the screen ($\chi^2$ $p = 1.1^{-7}$), and six as hits in the expected negative arm of the screen ($\chi^2$ $p = 0.01$). These are highlighted in Fig. 3b, where the fragments described above are ranked according to fold-change following either positive or negative selection (Fig. 3b); the highly repetitive *VSGs* and *ESAGs* were not surveyed as part of this analysis. We also employed the Wilcoxon rank-sum test to determine statistical significance and adjusted the resulting $p$-values for multiple comparisons using the False Discovery Rate (FDR). This assessment revealed significant enrichment (FDR < 0.1) for 1112 positive regulatory hit fragments from 1070 3'-UTRs, and 807 negative regulatory hit fragments from 801 3'-UTRs, (Supplementary Data 1).

The known regulatory 3'-UTRs detailed above, that were also associated with significantly enriched fragments in the MPRA, serve as exemplars, and for a sub-set of these, we show both MPRA read-mapping and gene expression: including tracks for RNA-seq read-depth to indicate mRNA abundance[27], and ribosome profiling data to indicate translated protein-coding regions and translation efficiency[5]. First, positive regulatory hit-fragments are shown for the aldolase and phosphoglycerate kinase (PGK) genes (Fig. 3c). The aldolase 3'-UTR was the positive regulatory UTR used for initial validation of the reporter assay (Fig. 1a, b), while the PGKA-C locus illustrates differential control of adjacent paralogs. PGKC yields a hit-fragment following positive selection, consistent with specific, and 3'-UTR-dependent, upregulation in bloodstream-form cells[28]. Second, negative regulatory hit-fragments are shown for cytochrome oxidase subunit and nucleoside transporter genes (Fig. 3d). A cytochrome oxidase subunit 3'-UTR was used for initial validation of the reporter assay (Fig. 1a, b), while the nucleoside transporter locus illustrates three paralogs, all of which register hit-fragments, consistent with 3'-UTR-dependent negative regulation in nucleoside-replete conditions[15]. Third, the trypanosome hexose transporter (THT) locus

registers independent positive and negative regulatory hit-fragments (Fig. 3e), consistent with developmental regulation of both THT1 and THT2 paralogs[13]. Among these exemplars, several hit-fragments identify specific regulatory portions of each 3'-UTR (Fig. 3c–e).

The assessment above indicated that the MPRA effectively selected positive and negative regulatory *T. brucei* 3'-UTRs and UTR fragments at a genomic scale. A requirement for mRNA processing signals likely contributed to strand-bias following positive selection (Fig. 3a) and may also increase the proportion of hit fragments that include a downstream poly-pyrimidine tract and splicing site. Indeed, we found that 79% of significantly enriched blasticidin-selected hits and 59% of ganciclovir-selected hits included the native downstream splice site (Supplementary Data 1). Notably, an analysis of the cognate 3'-UTRs for these hits revealed significantly longer ($p = 4.5^{-12}$) UTRs associated with positive control relative to negative control (Fig. 3f).

Since *trans*-splicing-associated sequences have been reported to impact splicing efficiency[9], and splicing is coupled to polyadenylation of the upstream transcript, we wondered whether downstream splice-sites might impact expression of the reporter in our screen. We found no evidence in support of this view, however. First, poly-pyrimidine tracts associated with the processing of high or low abundance mRNAs do not appear to be substantially different (Supplementary Fig. 3a). Second, although the dinucleotides immediately preceding an AG splice-site occur at substantially different frequencies and are associated with differential abundance of cognate mRNAs (Supplementary Fig. 3b), we saw no evidence for selection bias for these sequences downstream of our UTR hit fragments, in either the positive or negative control screens (Supplementary Fig. 3c). We conclude that differences in native splice-sites, included downstream of the majority of our hit fragments, had little impact on expression of the reporter in our MPRA.

## Poly-purine tracts are enriched in the 3'-UTRs of highly translated mRNAs

To explore features that may drive differential expression, we trimmed significantly enriched hit fragments and retained only those reporter-adjacent regions that overlapped with 3'-UTRs for analysis (Supplementary Data 1). We first compared nucleobase composition and motif enrichment in positive and negative regulatory hit fragments from the MPRA. Positive regulatory fragments were more A-rich ($p = 1.1^{-22}$), C-poor ($p = 1.1^{-12}$), and U(T)-poor ($p = 5.9^{-5}$) relative to negative regulatory fragments; both sets of fragments were C-poor and U(T)-rich (Fig. 4a). Consistent with these differences in nucleobase composition, a motif search revealed highly significant enrichment of a 9-b, A-rich, poly-purine motif (at 4163 sites, 10.3 sites per kb, E value $1.7^{-73}$) in positive regulatory fragments (Fig. 4b).

Selection in our MPRA was contingent upon the activity of the reporter, such that 3'-UTRs affecting gene expression either at the level of mRNA abundance or translation impacted the output. To further explore the relationship between hits in the MPRA and gene expression control, we assessed previously published data for mRNA abundance, translation efficiency[5], and mRNA half-life[3], in relation to

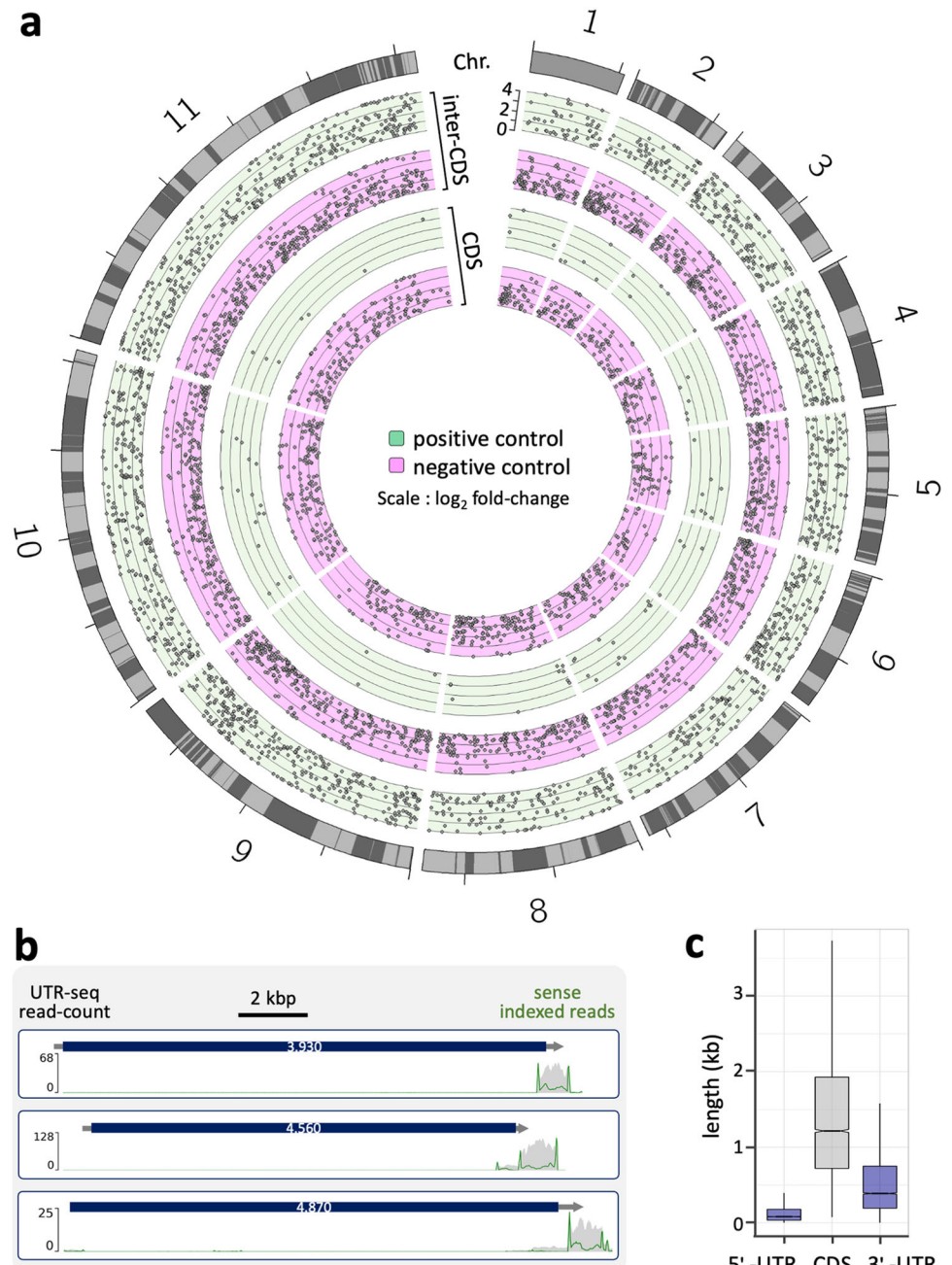

**Fig. 2 | Enrichment of 3'-UTR sequences following positive selection. a** The Circos plot shows an approx. 25 Mbp map of the *T. brucei* genome, incorporating eleven mega-base chromosomes, and encoding approx. 9000 genes. Polycistrons are indicated on the outer circle. Enrichment for DNA fragments inserted in the sense orientation in relation to the reporter, following blasticidin selection for positive control (green background), or ganciclovir selection for negative control (magenta background), is indicated. Scale is log$_2$-fold-change relative to plasmid control, with values clipped when > 4. **b** The maps show UTR-seq read-density for three exemplar genes. The grey lines with arrowheads indicate the UTRs and transcription from left to right. Sense paired, indexed reads highlight the boundaries of hit fragments, and are indicated by the green lines, with total reads indicated in grey. **c** The boxplot shows length data after updating the *T. brucei* 3'-UTR annotations. Boxes indicate the interquartile range (IQR), the whiskers show the range of values within 1.5*IQR and a horizontal line indicates the median. The notches represent the 95% confidence interval for each median. $n$ = 8115 (5'-UTR), 8258 (CDS), 8022 (3'-UTR).

hit-fragment enrichment in the MPRA (Fig. 4c, Supplementary Fig. 4a, b); translation efficiency is calculated by dividing ribosome footprint read-counts by mRNA read-counts for each CDS[5]. We found that hits in the MPRA were more strongly correlated with differences in translation efficiency (Fig. 4c, $p = 1.2^{-68}$) than they were with differences in mRNA abundance or mRNA half-life (Supplementary Fig. 4a, b, $p = 1^{-18}$ and $1.3^{-21}$, respectively). A similar analysis, but this time assessing mRNA abundance and translation efficiency in relation to frequency of the 9-b, A-rich motif identified in 3'-UTRs above (Fig. 4d,

Supplementary Fig. 4c, d), revealed a clear correlation with translation efficiency (Fig. 4d, $R^2$ = 0.28), a weak inverse correlation with mRNA abundance (Supplementary Fig. 4c, $R^2$ = 0.02) and a weak correlation with mRNA half-life (Supplementary Fig. 4d, $R^2$ = 0.07). An assessment of UTR length (see Fig. 3f) and A-rich motif density within 3'-UTRs indicated that both the density and dosage of poly-purine tracts correlated with translation efficiency (Fig. 4e).

To further illustrate base composition bias and A-rich poly-purine tract enrichment, we show four exemplar gene loci, representing

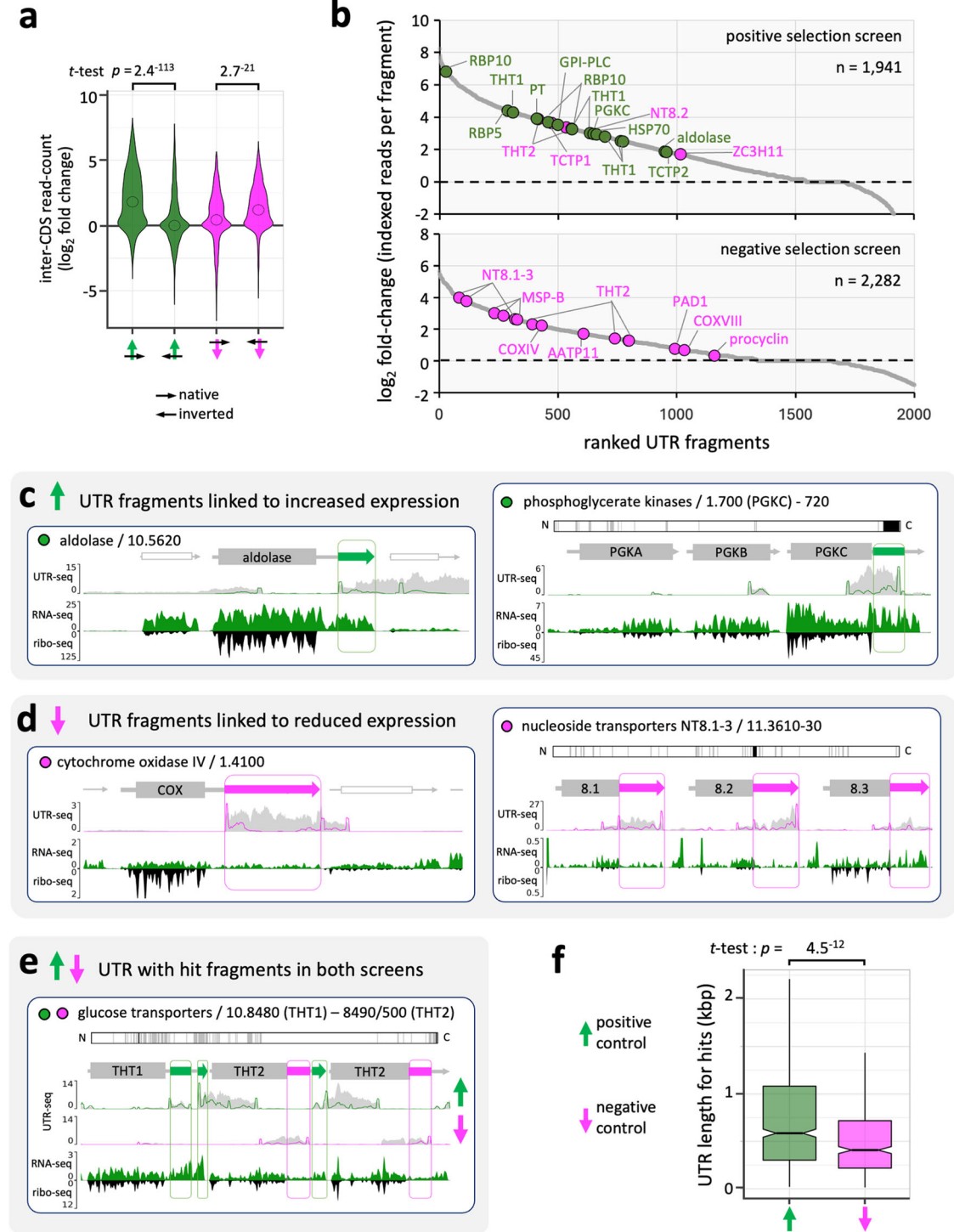

**Fig. 3 | Identification of thousands of regulatory 3'-UTRs. a** The violin plot shows relative enrichment of inter-CDS fragments cloned in their native or inverted orientation. Blasticidin selection for positive control; green, $n = 1941$ fragments. Ganciclovir selection for negative control; magenta, $n = 2282$ fragments. The open circles indicate median values, while $t$-tests were two-sided. **b** 3'-UTR associated fragments were ranked based on indexed read fold-change between the positive and negative selection screens. Previously published regulatory 3'-UTRs (see Supplementary Data 1) that feature as hits are highlighted. **c** The maps show UTR-seq read-density for two exemplar positive regulatory 3'-UTR fragments. Sense paired, indexed reads are indicated by the green lines, with total reads indicated in grey. UTR hit fragments are indicated as green boxes; with arrows indicating direction of transcription from left to right. RNA-seq (solid green) and ribosome profiling data (black) are also shown. The upper protein-map shows the relationship between the paralogs (PGKB-C) with identical amino acids shown in white, different amino acids in grey and additional segments in one of the proteins in black. **d** The maps show read-density for two exemplar negative regulatory 3'-UTRs. Sense indexed reads are indicated by the magenta line and UTR hit fragments are indicated as magenta boxed arrows. The paralogs compared in this case are NT8.1-2. Other details as for (**c**). **e** The map shows read-density for the hexose transporter locus with both positive and negative regulatory 3'-UTR fragments. Other details as for (**c**, **d**). **f** The boxplot shows length data for putative positive ($n = 844$) and negative ($n = 468$) regulatory 3'-UTRs associated with hit fragments that also include downstream mRNA processing sequences. Boxes indicate the interquartile range (IQR), the whiskers show the range of values within 1.5*IQR and a horizontal line indicates the median. The notches represent the 95% confidence interval for each median. The $t$-test was two-sided.

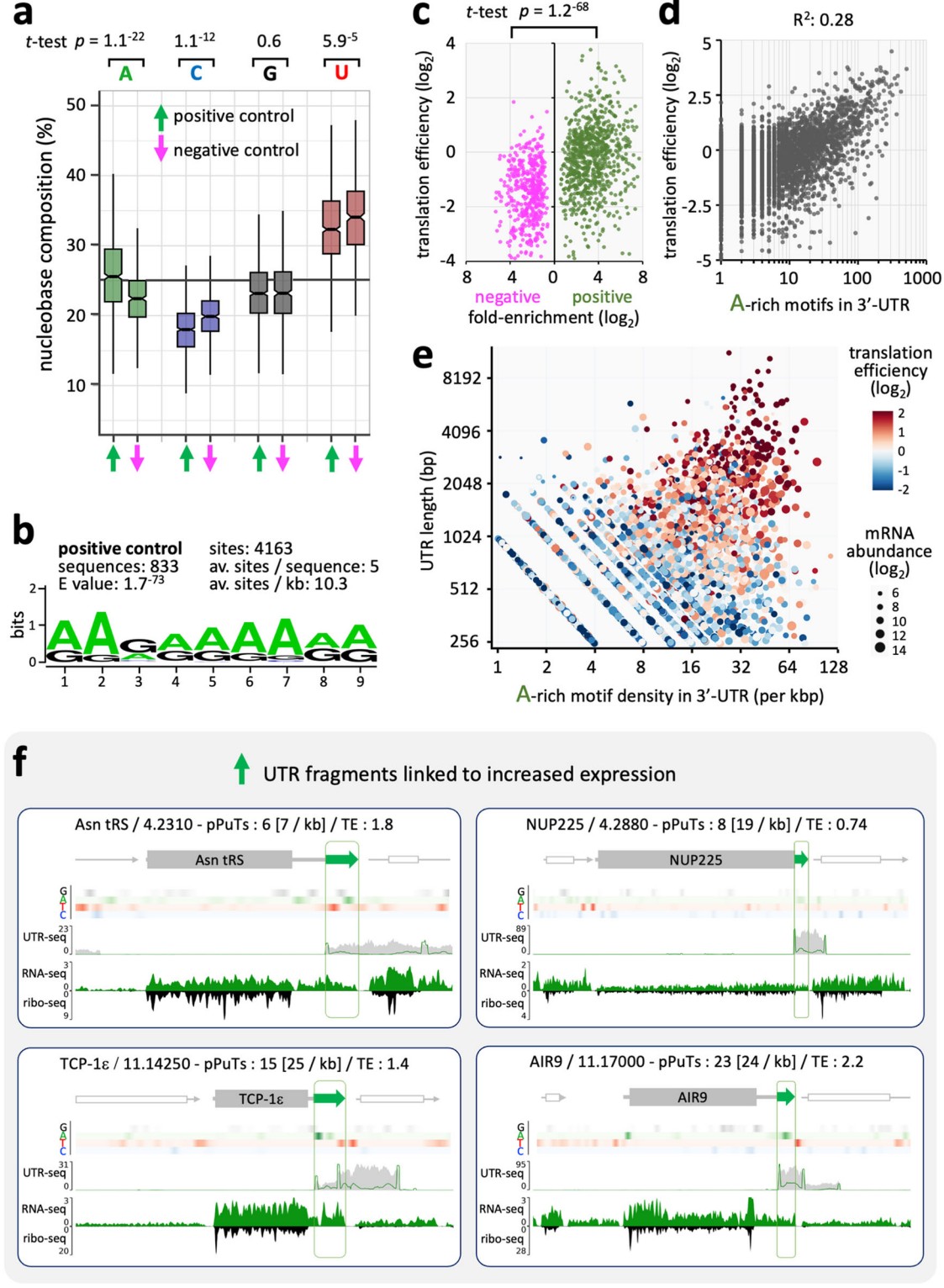

**Fig. 4 | Poly-purine tracts are enriched in the 3'-UTRs of highly translated mRNAs. a** The boxplot shows nucleobase composition for positive ($n = 833$) and negative ($n = 464$) regulatory hit fragments of > 20 b in length, that also include downstream mRNA processing sequences. Boxes indicate the IQR, the whiskers show the range of values within 1.5*IQR and a horizontal line indicates the median. The notches represent the 95% confidence interval for each median, while *t*-tests were two-sided. **b** The poly-purine tract (pPuT) motif shown was enriched in positive regulatory hit fragments relative to the negative fragments. **c** The plot shows fold-enrichment of hit fragments in the screen relative to published measures of translation efficiency[5]. The *t*-test was two-sided. **d** The plot shows number of A-rich motifs shown in b in 3'-UTRs relative to published measures of translation efficiency. $n = 4220$; FIMO settings '$p < 0.01$'. **e** The plot shows published measures of translation efficiency relative to density of A-rich motifs (FIMO setting '$p < 0.01$'), 3'-UTR length (all > 250 b), and mRNA abundance[5]. $n = 3608$. **f** The maps show UTR-seq read-density for four exemplar positive regulatory 3'-UTRs. Tracks showing nucleobase density are included. Other details as for Fig. 3c. Numbers of pPuT motifs in each UTR and translation efficiency (TE) measures are also indicated.

positive control hits in our screen, and transcripts with translation efficiency scores that are all above the median value of 0.55[5]. The gene maps display MPRA read-mapping and gene expression as above, and now also include tracks showing G-A-T-C base-composition (Fig. 4f). The nucleobase composition tracks effectively highlight low-complexity sequences, including T(U)-rich poly-pyrimidine tracts, known to be enriched in trypanosomatid 3'-UTRs[1]. Notably, A-rich regions can also be seen in each exemplar 3'-UTR and hit-fragment, while prominent A-rich regions are not seen in the 3'-UTRs of several flanking genes (Fig. 4f); the number of poly-purine motifs present in these four exemplar 3'-UTRs ranges from six to twenty-three. We conclude that A-rich poly-purine tracts are specifically enriched in the 3'-UTRs of highly translated mRNAs.

## Poly-purine tracts are enriched in the UTRs of highly translated paralogs

There are several examples of differentially expressed tandem paralogous genes in trypanosomatids, such as the phosphoglycerate kinase and hexose transporter genes shown in Fig. 3c, e. We wondered whether other examples would display A-rich regions and poly-purine motifs specifically enriched in the 3'-UTRs of a preferentially expressed paralog. A scan of hits recovered in the screen revealed differential poly-purine tract abundance in nine tandem paralog pairs (Fig. 5a). These genes encode mRNA-binding proteins[29–32], cytoskeletal proteins[33], transporters[34–36], nucleolar proteins[37], and hexokinases[38]. The nucleobase composition tracks highlight A-rich regions in all positive regulatory 3'-UTR hit fragments from the MPRA (Fig. 5a). Indeed, poly-purine tract density and 3'-UTR length are predictive of differential translation for all of these paralog-pairs, while three paralogous 3'-UTRs associated with lower expression completely lacked A-rich poly-purine tracts (Fig. 5b). A lesser expressed zinc transporter 3'-UTR also yielded a hit-fragment in the negative control arm of the MPRA (Fig. 5a). These results indicated that the MPRA not only identified thousands of regulatory sequences, but also effectively distinguished between the 3'-UTRs of otherwise closely related paralogs.

## Predicting translation efficiency using UTR sequences alone

The analyses above suggested that A-rich poly-purine tracts increase translation in *T. brucei*. We next used a machine learning approach to predict expression from UTR sequences alone as an orthogonal approach to the identification of *cis*-acting sequences. We considered both 5'-UTRs and 3'-UTRs for this analysis and first assessed nucleobase composition in relation to translation efficiency. Consistent with analysis of hit-fragments above (Fig. 4a), 3'-UTRs associated with higher translation efficiency tended to be relatively A-rich and C-poor, which was also found to be the case for 5'-UTRs (Fig. 6a).

Machine learning was next used to predict translation efficiency. A set of 46 features were used to train a random forest regressor and to run the predictions, yielding Pearson correlation coefficients of 0.62 (Mean Squared Error = 0.02) when using 3'-UTRs alone (Fig. 6b, upper panel), and 0.66 (MSE = 0.018) when using both 5'-UTRs and 3'-UTRs (Fig. 6c, upper panel). We conclude that the algorithm effectively predicted observed measures of translation efficiency using 3'-UTR sequences alone, and that 5'-UTR sequences likely also contribute to translation efficiency control. The value of the top eight features and their contributions to the predictions were visualised using SHAP (SHapley Additive exPlanation) values[39]. 3'-UTR-length, A-rich tracts and low C-count made the greatest contribution to predictions of high (indicated in red) translation efficiency (Fig. 6b, lower panel). When both 5'-UTRs and 3'-UTRs were considered, 3'-UTR-length, 3'-UTR A-tracts, 5'-UTR AU-count, and 5'-UTR A-count made the greatest contribution (Fig. 6c, lower panel). Thus, the MPRA, as well as machine learning based predictions, implicated A-rich poly-purine tracts as the primary 3'-UTR *cis*-acting sequences that promote translation.

## Poly-purine and -pyrimidine rich 3'-UTRs in bloodstream up-translated mRNAs

Our findings above indicated that translation is controlled in a pervasive manner by dosage and density of poly-purine tracts in 3'-UTRs. We next sought to identify sequences that impact developmental life-cycle stage specific expression patterns. Using previously published translation efficiency data[5], we searched for motifs enriched in the 3'-UTRs of transcripts with >5-fold increased translation in bloodstream-form cells, relative to insect-stage cells. This analysis revealed significant enrichment of a 9-b, U-rich, poly-pyrimidine tract (at 4.2 sites per kb, E value $1.1^{-26}$) (Fig. 7a). An assessment of both A-rich and U-rich motif density in 3'-UTRs associated with >4-fold developmentally regulated genes, in relation to either mRNA abundance or translation efficiency[5], revealed significantly higher densities of both the A-rich poly-purine motif shown in Fig. 4b, and the U-rich poly-pyrimidine motif shown in Fig. 7a, in 3'-UTRs associated with bloodstream-form up-translated mRNAs (Fig. 7b). Poly-pyrimidine tracts were also significantly enriched in 3'-UTRs associated with more abundant transcripts in insect-stage cells (Fig. 7b).

These findings suggest that a high density of poly-pyrimidine tracts reduces translation (and mRNA stability) in bloodstream-form trypanosomes, also mitigating the positive effect of poly-purine tracts on translation when both sequences co-occur at high density. An assessment of co-occurrence of these sequences in 3'-UTRs associated with >4-fold up-translated transcripts in either bloodstream-form or insect-stage cells yielded data consistent with this hypothesis (Fig. 7c); this cohort of bloodstream-form up-translated transcripts was specifically overrepresented ($\chi^2$ $p = 1.8^{-7}$) for longer (>2 kbp) 3'-UTRs with a high density (>20 per kbp) of both poly-purine and poly-pyrimidine tract motifs (see Fig. 7c, left-hand panels).

We identified forty-four >3-fold up-translated transcripts in bloodstream-form cells, with 3'-UTRs of >2 kbp and with >20 poly-purine and poly-pyrimidine tract motifs per kbp. Transcripts previously noted to have long 3'-UTRs[1], those associated with Gene Ontology terms for 'mRNA binding' ($p = 1.4^{-16}$), or 'protein kinase activity' ($p = 2.3^{-8}$), were overrepresented in this cohort. Notably, these include the bloodstream-specific *VSG* mRNA-binding protein, CFB2[29,32] (see Fig. 5a), the known 3'-UTR regulated mRNA-binding protein, RBP10 [12], and the repressor of differentiation kinase, RDK2[40]; both RBP10 and RDK2 are known to promote the bloodstream-form phenotype.

The relative density of A-rich and U-rich motifs, 3'-UTR length, and translation efficiency differential, are visualised for four exemplars of these bloodstream-form up-translated transcripts; RBP10, RDK2, the known 3'-UTR regulated glycosylphosphatidylinositol-specific phospholipase, GPI-PLC[41], and the bloodstream-specific alternative oxidase[42], and also for two insect-stage up-translated exemplars (Fig. 7c, right-hand panel). All 3'-UTRs for these genes yielded hit-fragments in the respective positive and negative control arms of the MPRA, and we show a map for each gene locus. The individual maps display MPRA read-mapping, and tracks showing G-A-T-C base-composition, as above, and now include gene expression tracks for both bloodstream-form and insect-stage cells[5] to highlight developmental differences (Fig. 7d, e). A high density of both A-rich and U-rich regions can be seen in the 3'-UTRs of bloodstream-form up-translated genes (Fig. 7d). Indeed, three independent hit-fragments were associated with RBP10, consistent with recent dissection of this >7 kb 3'-UTR[12]. In contrast, a high density of U-rich sequences can be seen in the 3'-UTRs of insect-stage up-translated genes (Fig. 7e), encoding the amino acid transporter[18], and major surface protease[21], both known to be regulated by their 3'-UTRs. Indeed, a U-rich element in the 3'-UTR appears to suppress translation in bloodstream-form cells in the latter case[21]. We conclude that long 3'-UTRs enriched in both poly-purine and poly-pyrimidine tracts often display increased translation in bloodstream-form cells.

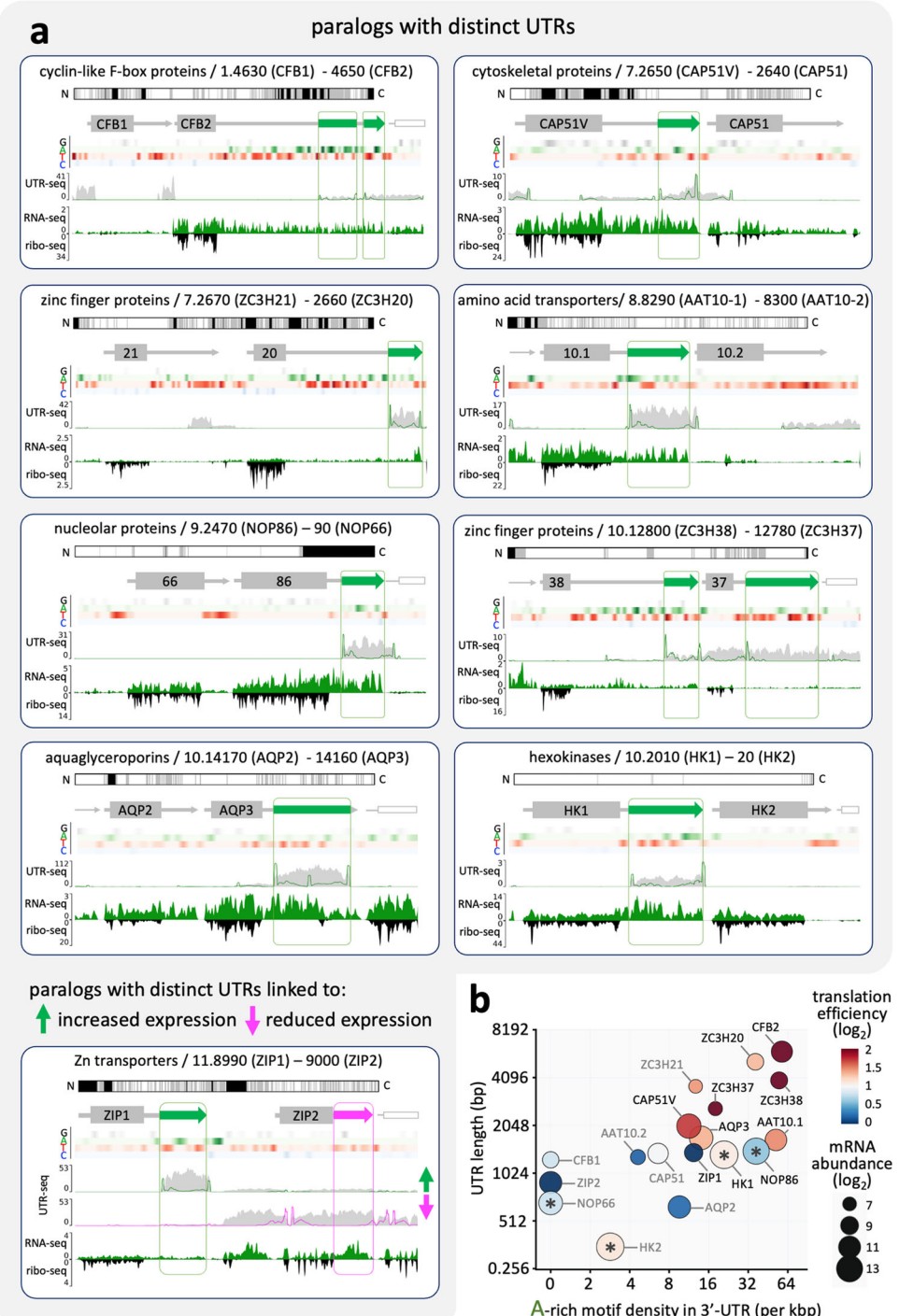

**Fig. 5 | Poly-purine tracts are enriched in the UTRs of highly expressed paralogs. a** The maps show UTR-seq read-density for nine tandem paralog pairs, all with positive regulatory 3'-UTR hit fragments, > 10 A-rich motifs (FIMO setting '*p* < 0.01') in the 3'-UTR of one paralog, and with < 33% the number of A-rich motifs in the other paralog. Tracks showing nucleobase density are included. Other details as for Fig. 3c, d. **b** The plot shows published measures of translation efficiency and mRNA abundance[5] relative to density of A-rich motifs (FIMO setting '*p* < 0.01') and 3'-UTR length for the paralog pairs in a. The darker text labels indicate those genes with hit-fragments in the positive selection screen. *, translation efficiency measures appear similar for NOP66/86 and for hexokinases because the coding sequences are largely identical. The 3'-UTRs are distinct, however, and reveal differential expression (**a**).

## Synthetic positive regulatory 3'-UTRs are enriched in poly-purine tracts

Our results above indicated a role for poly-purine tract dosage and density in increasing translation efficiency, and also suggested a role for poly-pyrimidine tracts in modulating poly-purine tract function. Indeed, an assessment of co-occurrence of these distinct motifs in 3'-UTRs indicated a tendency for enrichment of both sequences in long

UTRs (Fig. 8a). To further explore 3'-UTR sequence - function relationships, and as an additional test of our hypothesis that A-rich poly-purine tracts increase gene expression, we took advantage of library fragments cloned in reverse orientation relative to their native context, that were derived entirely from within native 3'-UTRs, and that were enriched following positive selection in our MPRA (see Fig. 8b). Forty-three such inverted and synthetic UTR fragments of >500 b were

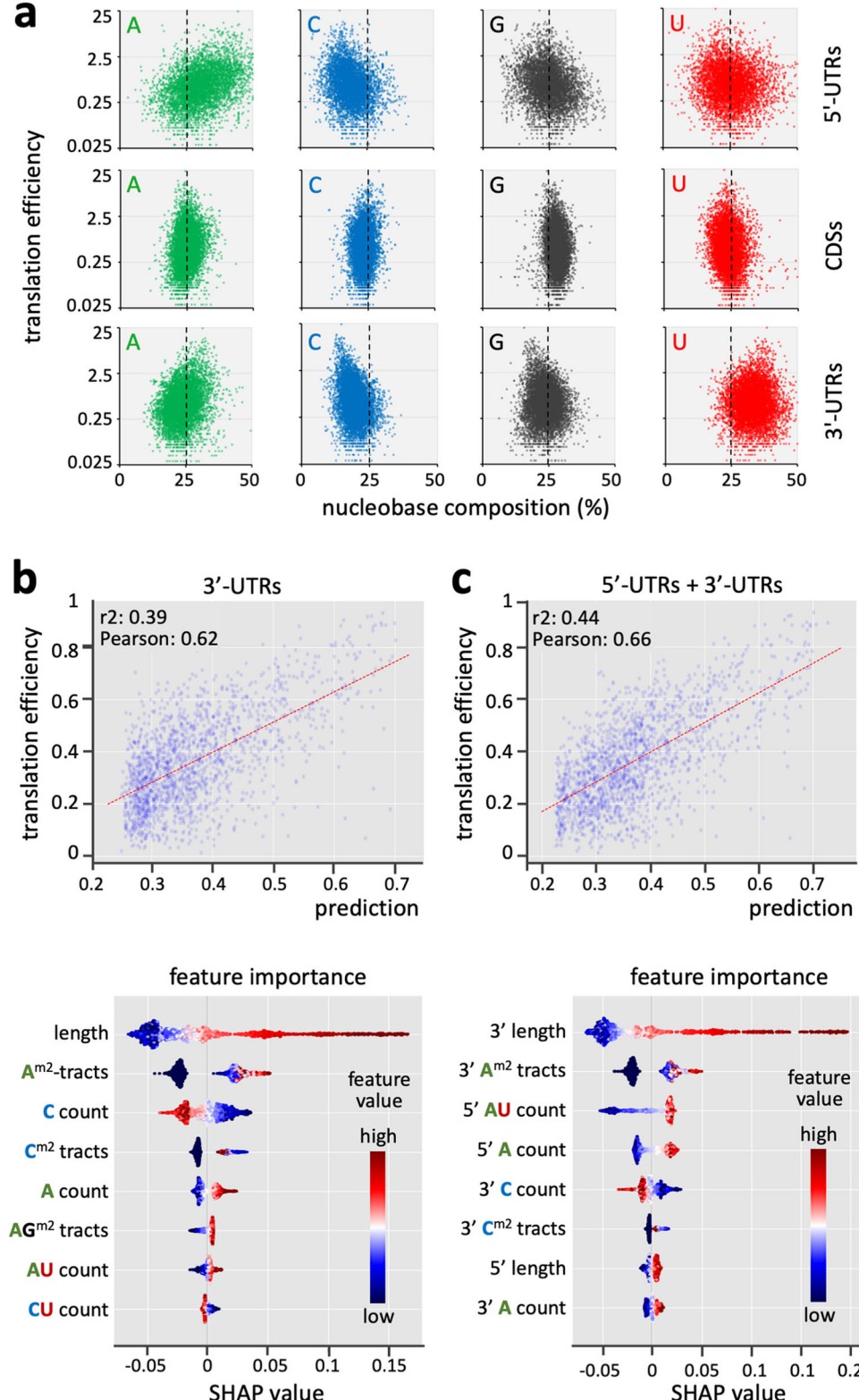

**Fig. 6 | Predicting translation efficiency using UTR sequences alone. a** The plots show nucleobase composition for 5′-UTRs of >49 b, CDSs of >199 b, and 3′-UTRs of > 99 b, relative to published measures of translation efficiency[5]. **b** Machine learning model evaluation based on 3′-UTR sequences. The upper plot shows translation efficiency values for the test set (*n* = 2020 genes) compared with the model predictions. A linear regression line is shown. The lower plot shows the SHAP values for each gene and for the top eight features that contribute to the predictions. The

colour scale reflects relative contribution to high (red) or low (blue) translation efficiency. The dots are jittered in the *y*-axis to illustrate the distribution of the SHAP values. $A^{m2}$, A-tracts longer than 5 allowing 2 mismatches; $C^{m2}$, C-tracts longer than 5 allowing 2 mismatches; $AG^{m2}$, AG-tracts longer than 5 allowing 2 mismatches. **c** Machine learning model evaluation based on 5′-UTR and 3′-UTR sequences. *n* = 2016 genes. Other details as in panel b.

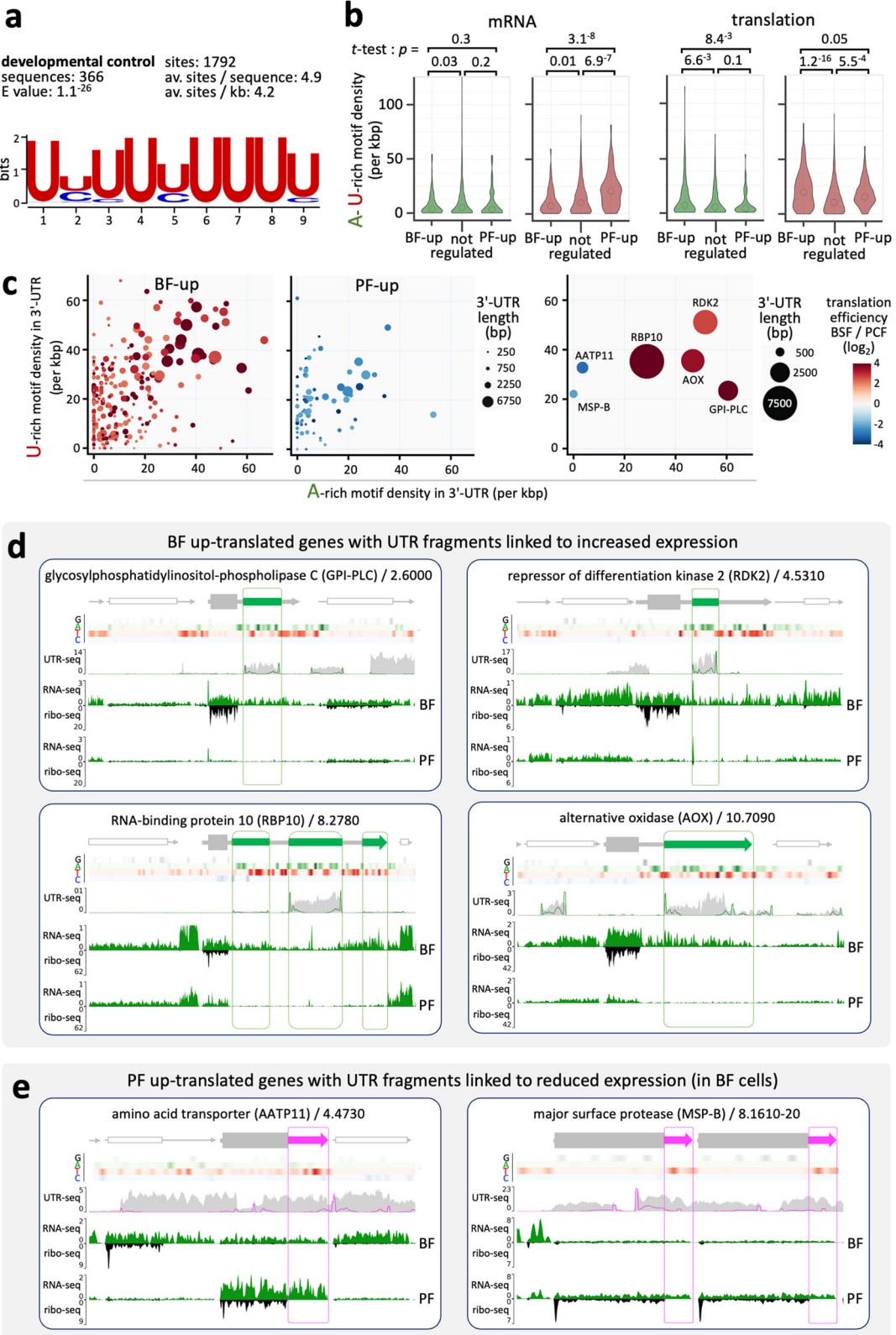

**d** BF up-translated genes with UTR fragments linked to increased expression

**e** PF up-translated genes with UTR fragments linked to reduced expression (in BF cells)

found to be significantly enriched following blasticidin-selection. These fragments displayed an A-rich motif density that was on average 3.3-times greater that the U-rich motif density (Fig. 8c). Thus, T(U)-rich poly-pyrimidine tracts, often found in *T. brucei* 3′-UTRs[1], increased reporter expression when reoriented to yield synthetic A-rich poly-purine tract-rich 3′-UTRs, thereby mimicking native positive regulation. Given similar proportions of U-rich and A-rich motifs in longer

3′-UTRs, we conclude that there was substantial selection for, and enrichment of, A-rich sequences in this synthetic arm of the MPRA. We mapped and visualized both poly-purine and -polypyrimidine motif distribution in these hit fragments (Fig. 8d). Notably, in addition to densely packed A-rich poly-purine motifs, synthetic U-rich poly-pyrimidine motifs likely served as synthetic downstream splicing signals, similar to those 'cryptic splice sites' described previously[8]. Thus,

**Fig. 7 | Poly-purine and -pyrimidine rich 3′-UTRs in bloodstream up-translated mRNAs. a** The motif shown was enriched in 3′-UTRs that displayed >5-fold upregulated translation[5] in bloodstream-form cells relative to control UTRs (<10% difference between life-cycle stages; n = 1450). **b** The violin plots on the left show motif density for 3′-UTRs of >250 b that displayed >4-fold upregulated mRNA abundance in bloodstream-form cells (BF-up; n = 139), or insect-stage, procyclic form cells (PF-up; n = 73), relative to control UTRs (<10% difference between life-cycle stages; n = 999). The violin plots on the right show motif density for 3′-UTRs of >250 b that displayed >4-fold upregulated translation in bloodstream-form cells (BF-up; n = 278), or insect-stage, procyclic form cells (PF-up; n = 81), relative to control UTRs (<10% difference between life-cycle stages; n = 818). Data are shown for the A-rich poly-purine motif in Fig. 4b, and for the U-rich poly-pyrimidine motif in Fig. 7a, (FIMO settings 'p < 0.01'). Open circles indicate median values while *t*-tests were one-sided. **c** The plots show density of A-rich motifs and U-rich motifs (FIMO settings 'p < 0.01'), and 3′-UTR length and also published ratios of translation efficiency[5] for the bloodstream form up-translated and procyclic form up-translated sets of transcripts described in (**b**). The plot on the right shows data for those six exemplar 3′-UTRs detailed in (**d, e**), four from the BF-up set and two from the PF-up set. **d** The maps show UTR-seq read-density for four exemplar 3′-UTRs associated with bloodstream form up-translated genes and with positive regulatory hit fragments in the screen. Tracks showing nucleobase density are included. Other details as for Fig. 3c. **e** The maps show UTR-seq read-density for two exemplar 3′-UTRs associated with insect-stage up-translated genes and with negative regulatory hit fragments in the screen. Tracks showing nucleobase density are included; other details as for Fig. 3d.

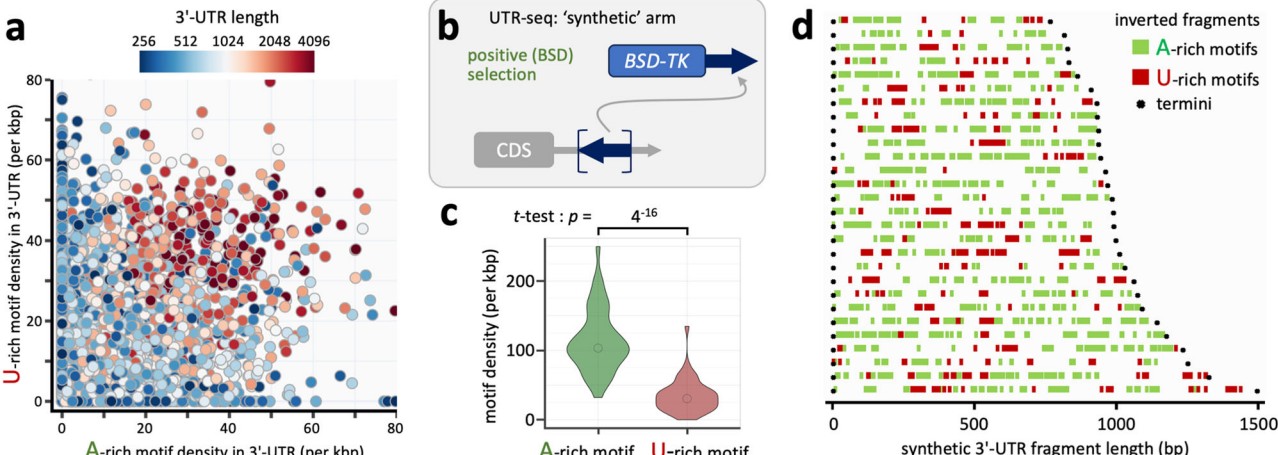

**Fig. 8 | Synthetic positive regulatory 3′-UTRs are enriched in poly-purine tracts. a** The plot shows 3′-UTR length relative to density of A-rich motifs and U-rich motifs (FIMO settings 'p < 0.01'). n = 4891; UTRs of > 250 b. **b** The schematic shows assessment of hit fragments in the positive selection arm of the MPRA that were derived from native 3′-UTRs, but inverted relative to their native orientation. The *t*-test was one-sided. **c** The violin plot shows motif density for all significantly enriched synthetic fragments of > 500 b. n = 43. Data are shown for the A-rich poly-purine motif in Fig. 4b, and for the U-rich poly-pyrimidine motif in Fig. 7a (FIMO settings 'p < 0.01'). Open circles indicate median values. **d** The maps show the distribution of A-rich and U-rich motifs in those hit fragments of between 750 and 1500 b in length.

analysis of synthetic, positive regulatory 3′-UTRs in the MPRA further supported the robust correspondence between A-rich poly-purine tracts and positive control.

## Discussion

Eukaryotic cells express thousands of proteins that differ in abundance over a wide range, and regulation involves interactions between mRNA 3′-UTRs and mRNA binding proteins. *T. brucei* is an important protozoan parasite and, largely due to pervasive polycistronic transcription, a model organism for studies on post-transcriptional gene expression controls. Knowledge regarding the mechanisms involved remains limited, however. We devised a massive parallel reporter assay coupled to genome-scale UTR-seq to identify *cis*-regulatory sequences embedded within *T. brucei* 3′-UTRs. Analysis of thousands of regulatory 3′-UTRs and UTR fragments, and a machine learning approach, revealed a *cis*-regulatory code underpinning differential expression control.

Our findings indicate that poly-purine tracts in 3′-UTRs drive increased translation in a dosage- and density-dependent manner. Although our reporter assay was not designed to assess developmental controls, our analysis also suggests that poly-pyrimidine tracts can conditionally modulate the activity of poly-purine tracts. More specifically, we propose recruitment of translation machinery by positive regulatory A-rich poly-purine tracts (Fig. 9a), and base-pairing mediated masking of these sequences by U-rich poly-pyrimidine tracts in the same transcript (Fig. 9b). We suggest that conditional unmasking of poly-purine tracts contributes to increased expression in bloodstream-form cells, potentially driven by temperature differences encountered in the mammalian bloodstream or insect vector.

Limitations of our study include a readout from our reporter assay that depended upon protein product, meaning that impacts of a 3′-UTR on mRNA maturation, translation, and turnover were not assessed separately, as well as lack of independent validation of novel hit fragments and 3′-UTRs. Our reporter assay was also conducted using only bloodstream form cells, meaning that our hypothesis regarding developmental control through (un)masking of poly-purine tracts is more speculative. Future studies in this area should address these limitations and may reveal regulatory factors that control expression by interacting with poly-purine and/or poly-pyrimidine tracts, and/or other regulatory sequences, in 5′-UTRs and/or 3′-UTR.

mRNA-binding proteins are undoubtedly involved in decoding the *cis*-regulatory code we describe here, and typanosomatids express large numbers of predicted mRNA-binding proteins[43]. Indeed, a genome-wide tethering screen identified approx. 300 *T. brucei* proteins that control gene expression when bound to a reporter mRNA[44]. Notably, and consistent with our findings, *T. brucei* transcripts with U-rich 3′-UTRs are unstable and degraded by the nuclear exosome[45,46]. RBP10 also binds insect-stage specific transcripts containing UA(U)$_6$ motifs, reducing translation and increasing turnover in bloodstream form cells[47], while another mRNA-binding protein, DRBD18, impacts polyadenylation and facilitates the production of longer mRNAs with longer 3′-UTR[48–50]. Other bespoke regulatory sequences are undoubtedly also embedded within *T. brucei* UTRs, including AU-rich elements

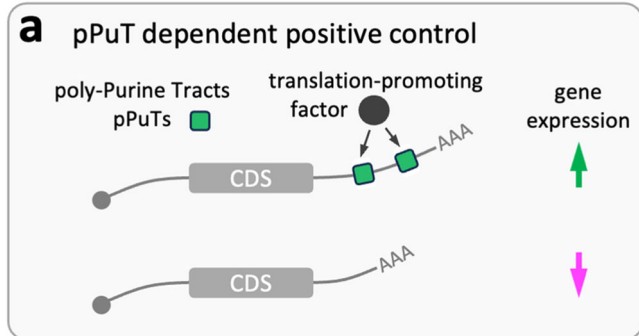

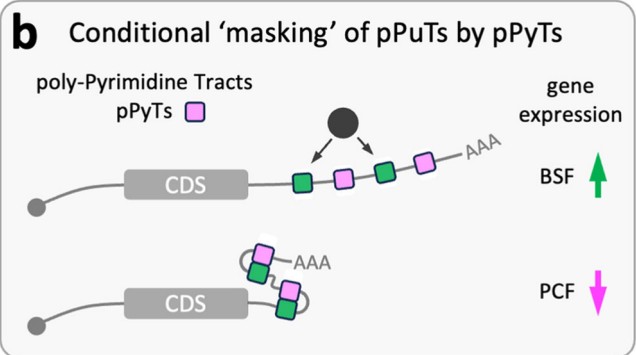

**Fig. 9 | A model for UTR-based post-transcriptional expression control in *T. brucei*. a** Poly-purine tracts (pPuTs) in 3'-UTRs drive increased translation in a dosage- and density-dependent manner. **b** pPuTs and poly-pyrimidine tracts (pPyTs) in 3'-UTRs interact such that pPyTs mask pPuTs and reduce gene expression in procyclic form (PF, insect stage) cells. Changes in secondary structure, possibly due to temperature differences, could be responsible for pPuT (un) masking. 5'-UTR sequences may behave similarly.

in 3'-UTRs, implicated in developmental controls involving transcript destabilisation by RBP6[51], or stabilisation by either ZC3H11[52] or DHH1[53]. Other examples include *VSG* transcript binding and positive control by CFB2[32], and procyclin transcript binding and negative control by NRG1[54]. In terms of specific responses to environmental cues, PuREBP1-2 negatively regulates nucleoside transporter transcripts by binding a stem-loop structure in the 3'-UTR[55], and RBP5 binds and regulates its own transcript in an iron-responsive manner[19]. The sequences involved in recruiting other mRNA binding proteins that orchestrate key developmental transitions, such as ZFP1[56], ZFP3[57], RBP7[58] and ZC3H20[30], are less clear. Modulation of RNA folding, structure, and function may also be impacted by mRNA-binding proteins, or long non-coding RNAs[59]. Our approach, annotation and dataset should facilitate future efforts to identify bespoke *cis*-acting UTR sequences and regulons in the trypanosomatids, and to determine how UTRs, as well as codon usage bias[2,4], impact translation and mRNA turnover.

Our findings suggest a remarkably versatile strategy for evolving new regulatory UTRs from low-complexity, repetitive and abundant sequence motifs. A-rich and T-rich tracts are abundant in the *T. brucei* genome and homologous recombination and microhomology-mediated end-joining are the dominant DNA repair pathways[60]. We envision a system that facilitates rapid adaptation of gene expression through assembly of novel UTRs. This mechanism may drive developmental stage-specific expression and/or differential expression of paralogs following gene duplication, and similar mechanisms may operate in other trypanosomatids. Indeed, we find several examples of tandem paralogous genes with distinct UTRs in *T. brucei*. We also show that inverted genomic U(T)-rich sequences form 'synthetic' A-rich positive regulatory UTRs in our screen. Notably, poly-purine tracts are over-represented in the genomes of other eukaryotes, including yeast

and humans[61]. Indeed, gene expression is primarily controlled at the level of translation in human cells[62], where purine-rich 5'-UTRs can promote translation factor binding and translation[63], suggesting that our findings could have broader implications. In conclusion, we describe post-transcriptional reprogramming underpinned by a *cis*-regulatory code embedded within trypanosome UTRs.

## Methods

### *T. brucei* growth and manipulation
Bloodstream form Lister 427 *Trypanosoma brucei* and derivatives, including 2T1[64], were cultured in HMI-11 (Gibco) supplemented with 10% fetal bovine serum (Sigma) at 37 °C and with 5% $CO_2$ in a humidified incubator. Genetic manipulations were carried out by electroporation using a Nucleofector (Lonza), with cytomix for routine transfections and a Nucleofector Human T-cell kit (Lonza) for high efficiency library transfections[65]. Strains expressing thymidine kinase were cultured in HMI-11 lacking thymidine, made from IMDM base media (ThermoFisher Scientific) and supplemented with 1 mM hypoxanthine (Sigma), 0.05 mM bathocuprione disulphonic acid (Sigma), 1 mM sodium pyruvate (Sigma), 1.5 mM L-cysteine (Sigma) and 10% fetal bovine serum (Sigma).

### pRPaiUTR plasmid construction
A LacZ stuffer fragment was amplified using primers UTR5 and UTR3 (Supplementary Data 1) and cloned into pRPaiSL [66] using ApaI and KpnI restriction sites; this added flanking BbsI and FseI sites to the LacZ stuffer to facilitate library assembly. Next, a synthetic DNA fragment (GeneArt) consisting of the tubulin 5'-UTR and a blasticidin-Ty1-thymidine kinase fusion cassette was cloned upstream of the LacZ stuffer using BamHI and AflII restriction sites. A second synthetic DNA fragment (GeneArt) containing a bloodstream *VSG* expression site (ES) promoter, aldolase 5'-UTR, neomycin phosphotransferase gene and actin 3'-UTR was then cloned between the rDNA promoter and hygromycin targeting region using NheI and NdeI restriction sites. The aldolase 3'-UTR fragment (695 bp) was amplified by PCR using the Ald3UTRFseIF and Ald3UTRFseIR primers and the COXV 3'-UTR fragment (204 bp) was amplified using the CoxV3UTRFseIF and Cox-V3UTRFseIR primers (Supplementary Data 1). The PCR products were cloned into pRPaiUTR downstream of the BSD-TK reporter gene and in place of the LacZ stuffer. The resulting pRPaiUTR constructs were digested with AscI prior to transfection into 2T1 cells.

### Dose-response assays
Cells were plated in 96-well plates at $1 \times 10^3$ cells/ml in a 2-fold serial dilution of selective drug; blasticidin (Melford) or ganciclovir (Sigma). Plates were incubated at 37 °C for 72 h, 20 µl resazurin sodium salt (AlamarBlue, Sigma) at 0.49 mM in PBS was added to each well and plates were incubated for a further 6 h. Fluorescence was determined using an Infinite 200 pro plate reader (Tecan) at an excitation wavelength of 540 nm and an emission wavelength of 590 nm. Data were analysed and $EC_{50}$ values were derived using Prism (GraphPad).

### Protein blotting
*T. brucei* cell lysates were separated on a 10% SDS polyacrylamide gel and transferred to nitrocellulose membrane (Protran, Amersham) using the semi-dry Turbo-blot system (Bio-Rad). Membranes were probed with anti-Ty1 mouse monoclonal antisera (Sigma) at 1:5000 followed by anti-mouse HRP-coupled secondary antisera (Bio-rad) at 1:10,000. Signal was visualised using a chemiluminescence kit (Amersham) according to the manufacturer's instructions.

### UTR plasmid library and *T. brucei* library assembly
The UTR library was assembled essentially as described previously[67], except that the genomic fragments used were 1–3 kbp in size and they were cloned into pRPaiUTR. Briefly, pRPaiUTR was digested with BbsI and

semi-filled with dTTP using Klenow DNA polymerase I (NEB). *T. brucei* genomic DNA was partially digested with Sau3AI (NEB) for 1 h at 37 °C. The DNA was separated on a 1% agarose gel and DNA corresponding to 1–3 kbp in size was excised and extracted using a gel extraction kit (Qiagen). The gDNA fragments were semi-filled with dGTP as above and then ligated to semi-filled pRPaiUTR using T4 DNA ligase (NEB) overnight at 16 °C before electroporation into MegaX DH10B Electrocomp T1R cells (ThermoFisher). Dilutions from a 500 mL culture were grown on plates, and plasmids from 30 colonies were analysed following FseI digestion. The plasmid library was then isolated using a HiSpeed Plasmid Maxi Kit (Qiagen) and digested with AscI prior to nucleofection into *T. brucei*. Briefly, puromycin sensitive 2T1-Sce* cells were generated[65] and induced with tetracycline (1 μg/ml) for 8 h prior to nucleofection, which was carried out in 12 replicates using 12.5 μg digested UTR plasmid library DNA and $5 \times 10^7$ cells for each replicate. Selection (1 μg/ml phleomycin and 2 μg/ml G418) was applied 6 h later.

### UTR library screening
Sixty hours after library nucleofection, expression of the BSD-TK reporter cassette was induced by the addition of tetracycline (1 μg/ml) for 24 h. Either blasticidin (0.2 mg/ml) or ganciclovir (0.15 μg/ml) drug selection was applied, and growth was monitored using a haemocytometer over 8 days. Each arm of the screen was initiated with $4 \times 10^7$ cells and cell populations were maintained at a minimum of $2 \times 10^7$ throughout to maintain library complexity. Cells were harvested on days 4, 6 and 8, and genomic DNA was extracted using a DNeasy blood and tissue kit (Qiagen). pRPaiUTR library fragments were amplified from gDNA samples and the plasmid library using the pRPaUTRseq2 primer (Supplementary Data 1) and LongAmp polymerase (NEB) - 94 °C 5 min; 28 cycles: 94 °C 45 s, 50 °C 45 s, 65 °C 3 min 30 s; 65 °C 10 min. PCR amplicons were purified using a PCR purification kit (Qiagen).

### Annotation of 3'-UTRs
We refined and revised the 3'-UTR annotations for the *T. brucei* TREU927 core genome using the Apollo community annotation platform, available via TriTrypDB[26]. This re-annotation process was informed by nanopore RNA sequencing data[59]. We acquired the fast5 datasets (ERR7889821 and ERR7889820) and carried out base-calling with Guppy software (version 6.2.1 + 6588110, Oxford Nanopore Technologies) on a GPU infrastructure. The base-calling parameters were set with the flowcell specified as FLO-MIN106 and the kit as SQK-RNA002. We then employed Minimap2 (version 2.24-r1122) to align raw reads to the genome sequence, storing the alignments as BAM files. We also downloaded and re-aligned all available datasets from the SRA bioproject PRJNA634997[68]. A custom Python script was developed to filter BAM alignments and retain only those reads containing either the conserved 5' spliced leader sequence (TCTGTACTATATTG) or poly-A sequences of ten or more nucleotides in length. Coverage maps were generated from the BAM files using BamCoverage (version 3.5.0) from the DeepTools suite, with a resolution of 5-bp per window. To enhance the accuracy of 3'-UTR annotations in the *T. brucei* 927 reference genome, coverage maps and processed BAM files were uploaded into the Apollo web interface. This integration facilitated the manual revision of 4703 3'-UTR annotations across the organism's 11 main chromosomes. During this process, 3'-UTRs were adjusted — extended, shortened, or appended where previously unannotated — to align with the empirical data presented within the Apollo interface.

### UTR library sequencing and analysis
PCR amplicons from above were submitted for sequencing at the Tayside Centre for Genomic Analysis (University of Dundee). Sequencing libraries were prepared using the Nextera Flex kit (Illumina). Briefly, PCR amplicons were subjected to tagmentation to produce 300–350 bp fragments with tag and adapter sequences, and then amplified by limited-cycle PCR to add index adapters. The libraries were then pooled and sequenced on a NextSeq 500 (Illumina) platform to obtain 144–228 million 75-b paired-end reads per sample. Reads were aligned to the reference *T. brucei* genome v46, clone TREU927 from TriTrypDB[26] using Bowtie2[69], with the 'very-sensitive-local' pre-set alignment option. The alignments were converted to BAM format, reference sorted and indexed with SAMtools[70]. Alignment statistics were retrieved using the SAMtools stats function. A custom python script was used to separate, and save as BAM files, alignments containing the 5' ('CTGACTCCTTAAGGGCC') or the 3' ('GCCGGCCTCAGTTA') index sequences in either forward or reverse complement orientation. We used the peak detection algorithm, MACS2[71], to define enriched regions (peaks) in the blasticidin or ganciclovir selected samples relative to the plasmid library control sample. The parameters of MACS2 were --min-length 300 --max-gap 1 --broad --nomodel --keep-dup all -q 1 --broad-cutoff 1 --broad --llocal 3000. We then manually curated the boundaries of fragments that overlapped with annotated 3' UTRs, using indexed read-counts to record genomic coordinates. This was performed using the Apollo community annotation platform tool available at TriTrypDB. These curated fragments were converted to Simplified Annotation Format (SAF) and used to derive paired read-counts. The counts were determined using featureCounts from the Subread package[72] using the BAM files containing all the alignments and the BAM files containing only the index sequence specific alignments. The featureCounts parameters were: -p (pair-end) -B (both ends successfully aligned) -C (skip fragments that have their two ends aligned to different chromosome) -M (count multi-mapping) -O (match overlapping features). Using BEDtools intersect function[73], we assigned and trimmed blasticidin and ganciclovir selected fragments to align with the corresponding 3'-UTR sequences. Where a single fragment spanned across multiple 3'-UTRs, we selected the 3'-UTR overlapping segment closest to the *BSD-TK* CDS in the reporter construct.

### Principal component analysis
3'-UTR coordinates from *T. brucei* genome v46 were formatted into SAF for counting using featureCounts[72] and the BAM files containing all the alignments. The read counts were normalized using the StandardScaler function in the Python scikit-learn package to ensure comparability. Subsequently, principal component analysis was performed using the function in the Python scikit-learn package (https://scikit-learn.org/).

### Statistical analysis
Manually curated genomic fragments were used to count indexed reads from blasticidin and ganciclovir selected BAM files, yielding data for fragments cloned in the reporter construct in either native orientation relative to transcription, or in reverse orientation. Total read-counts were utilized to normalize indexed read-counts across samples, which were scaled by a factor of 10 million. The resulting values (paired reads per 10 million mapped paired reads) were rounded to the nearest integer. These read-counts were then compared pairwise between day-4, day-6 and day-8 blasticidin and ganciclovir selected samples, and indexed reads from opposite ends of each fragment were considered separately, yielding six pairwise comparisons. These values served as the basis for the two-sided Wilcoxon rank-sum test, for computing $\log_2$ fold changes and $\log_2$ average intensity; 1 was added to the datasets to avoid zero division for the latter two calculations. *p*-values obtained from the Wilcoxon test were adjusted for multiple comparisons using the False Discovery Rate (FDR) Benjamini-Hochberg (BH) method, implemented using the multipletests function from the statsmodels Python package (https://github.com/statsmodels/statsmodels/).

### Data visualisations
To visualize indexed read-count coverage around Sau3AI sites, we first utilized a custom Python script that employed regular expressions to

identify and annotate these recognition sites (GATC) in the megabase chromosomes of the TREU927 reference genome. The sites were recorded in BED format. We then consolidated BAM files containing paired reads with indexed sequences from blasticidin and ganciclovir-selected samples and the plasmid library control sample. Using the bamCoverage tool from the deepTools package[74], we computed read coverage at 1 bp resolution for all samples and saved these outputs in BigWig format. Subsequently, the computeMatrix function from the deepTools package was used to generate a count matrix for a 500 bp region around the Sau3AI sites, with settings to sum values across bins and treat missing data as zero. Finally, we visualized the resulting data as heatmaps using the plotHeatmap function from the deepTools package, sorting regions in descending order. For the Circos plots, fold-changes in indexed reads in the blasticidin or ganciclovir enriched regions relative to the plasmid library control were computed. Briefly, we used the indexed read-counts in forward orientation to compute fold-change for sense fragments and indexed read-counts in reverse orientation to compute fold-change for anti-sense fragments. The polycistronic regions were derived with a custom python script using the GFF file retrieved from TriTrypDB[26]. Briefly, strand annotation for gene-IDs were used to determine the orientation and length of the polycistronic regions. Any changes in gene orientation between the previous and next gene determine the starts of a new polycistronic region. The polycistronic regions were converted in karyotype annotation file for visualization in Circos[75] along with fold-change in read-abundance in blasticidin or ganciclovir enriched peaks relative to the plasmid library control. To visualise UTR-seq mapping for individual genes, genome coverage of aligned reads was extracted from the BAM files with BEDtools[73] (-bg option) to output bedGraph files, or with deepTools (bamCoverage)[74] to output bigWig files. The G-A-T-C frequencies were computed with a custom script using a window of 100 bp and saved as bedGraph files. The bedGraph files were converted to BigWig using the bedGraphToBigWig program[76]. The track files were visualized either with PyGraphviz (https://github.com/pygraphviz/pygraphviz) or svist4-get python packages[77]. Tracks showing RNA-seq data and ribosome profiling data[5] (cultured bloodstream form) were also included.

### Machine learning

A custom Python script was used to filter *T. brucei* 3'-UTRs, initially numbering 8261. The first step involved removal of 207 sequences of < 20 b. We further refined our dataset by removing 2027 sequences associated with genes exhibiting stage-specific expression, those with an iBAQ score < 4 in bloodstream form (BSF) and >4 in procyclic form (PCF), and vice versa[78]. The refined pool of 6214 UTRs was clustered using mmseqs, grouping the first 200 b of each sequence, and choosing representative sequences with homology < 0.4, yielding 5871 sequences for further analysis. Feature extraction from 5' and 3'-UTRs was performed using another Python script that harnessed the 're' module for regular expression pattern matching and Biopython for sequence manipulation[79]. Our script analyzed various DNA sequence attributes, such as nucleotide base counts and the prevalence of poly-purine, poly-pyrimidine, and homopolymeric stretches. These features were normalized against the total length of each 3'-UTR. To derive translation efficiency (TE) metrics for each gene, we used published ribosome profiling data[5], taking the ribo-seq replicate average and normalizing it against the combined average of ribo-seq and total RNA-seq (cultured bloodstream form), yielding a proportion between 0 and 1, yielding, for 5777 genes, TE values as a proportion between 0 and 1. We then assessed the predictive capability of 42 extracted features using a Random Forest Regressor from the scikit-learn library (https://scikit-learn.org/). Our dataset was partitioned into a 70% training set and a 30% test set. Features with Pearson correlation coefficients exceeding 0.95 were deemed redundant and removed, yielding 23 predictive features. The model's efficacy was evaluated using linear regression analyses conducted via the SciPy package[80].

Finally, to interpret the Random Forest model, we visualized the importance of the top eight features using SHAP values from the corresponding Python package, thereby elucidating the contribution of each UTR feature to the TE predictions[39].

### Sequence motif and base composition analysis

MEME (https://meme-suite.org/meme/tools/meme) was used to search for enriched un-gapped sequence motifs in hit-fragments or UTRs relative to a control dataset using the settings: differential enrichment mode, any number of repetitions, search given strand only. FIMO (Find Individual Motif Occurrences, https://meme-suite.org/meme/tools/fimo) was used to search hit-fragments or UTRs for occurrences of the motifs identified using MEME using the settings: match *p*-value '< 0.01', scan given strand only. Motif visualisations were generated using WebLogo (https://weblogo.berkeley.edu) and base composition analysis was carried out using Excel.

### Reporting summary

Further information on research design is available in the Nature Portfolio Reporting Summary linked to this article.

## Data availability

The data supporting the findings of this study are available from the corresponding author upon request. High-throughput sequencing data generated for this study have been deposited at the Sequence Read Archive under primary accession number PRJNA1076082. Source data are provided with this paper in the Source Data file. Source data are provided with this paper.

## Code availability

Code generated for this study and package versions have been deposited at GitHub (https://github.com/mtinti/utr_bsf_code) and Zenodo (https://zenodo.org/doi/10.5281/zenodo.10636308)[81].

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

## Acknowledgements

We thank Andrew Cassidy from the Tayside Centre for Genomic Analysis for advice on Illumina sequencing. This work was funded by Wellcome Trust Investigator Awards (100320/Z/12/Z to D.H. and 217105/Z/19/Z to D.H.).

## Author contributions

The experiments were designed by A.T. and D.H. and carried out by A.T. Initial mapping of UTR-seq data was carried out by R.W. Subsequent mapping, computational analysis and data visualisations were performed by M.T. Data analyses were performed by A.T., M.T. and D.H. Funding was acquired by D.H. The work was supervised by D.H. The manuscript was written by A.T., M.T. and D.H.

## Competing interests

The authors declare no competing interests.
