## [Peer Review File · Nature Communications]

Post-transcriptional reprogramming by thousands of mRNA untranslated regions in trypanosomesREVIEWER COMMENTS

Reviewer #1 (Remarks to the Author):

In trypanosomes, transcription of protein coding genes is polycistronic with co-transcriptional processing to produce monocistronic mRNAs. There is no obvious wide scale arrangement of genes into functional operons and little or no evidence for regulation of individual genes at the transcriptional level. This places the emphasis on post-transcriptional regulation of mRNA half life and therefore abundance.

The identification of trans- and cis-acting factors that modulate mRNA half life has produced a small number of examples but has not been overly successful.

This manuscript describes a whole genome approach to identifying positive and negative cis-acting elements in bloodstream from *Trypanosoma brucei*. The experimental approach was to clone fragments of genomic DNA downstream of a BSD-TK open reading frame allowing selection with blasticidin for up regulation and with glanciclovir for down regulation. This allowed the identification of fragments of genomic DNA that either increased or decreased the expression of a reporter.

The first figure outlines the screen and shows the outcome as a plot of positive and negative enrichments relative to the library content are shown. Very few positive regulators came from ORFs relative to inter-ORFs and this is illustrated by the distribution of fragments in three dynein genes.

Figure 2 shows: the distribution of 5'UTR, ORF and 3'UTR lengths, the distribution of enrichments for the fragments identified as increasing or decreasing reporter expression highlighting fragments from genes known to be differentially regulated in bloodstream form trypanosomes, five exemplars and the length distribution of 3'UTRs.

Figure 3 shows the base composition bias in different regulatory fragments, the purine rich motif identified as enriched in fragments that increase expression, a set of exemplars showing the location of 'positive and negative regulatory 3'UTR fragments' (Is this the same as the genomic DNA 'hit' or have the 3'UTR sequences been extracted?). Paralogs with different regulation are then analysed and the frequency of the polypurine motif show to correlated with increased expression.

Figure 4 shows the contribution of various features of the mRNA to translational efficiency and show the the 5' and 3'UTRs can be used to give a good prediction of translational efficiency.

Figure 5 considers sequences in the 3'UTR of mRNAs that are unregulated in bloodstream relative to insect stage trypanosomes and shows that a U-rich motif is over-represented in mRNAs upregulated in bloodstream forms and shows 4 exemplars.

As a brief summary, the findings were:

1. High translational efficiency in BSFs was associated with dosage of adenine rich poly purine tracts
2. 3'UTRs associated with upregulated expression in BSFs also enriched in U-rich polypyrimidine tracts

Overall this is elegant work and no further experiments are required. However the authors should consider the following: in interpreting their data.

Comments

1. The library screen selected for genomic fragments that resulted in successful maturation (polyadenylation) of the reporter mRNA and any cis-elements that selectively increased or decreased expression. The sequences required for polyadenylation of mRNAs in trypanosomes are complex as it is a two stage process, first the downstream mRNA is trans-spliced and then cleavage and polyadenylation occurs around 100 to 200 nucleotides upstream of the trans-splice site and location does not appear to depend on any sequence motif. Thus, for any fragment to confer expression on the reporter it has to extend well beyond the polyA site of the mRNA. This complex requirement should produce a strand bias.

It has previously been shown that the efficiency of trans-splicing, and therefore polyadenylation of the reporter, is sequence dependent (PMID: 16227607) so the expression of the reporter will be affected by the efficiency of mRNA maturation and any 3'UTR elements.

2. In several places I found it hard to distinguish in the text whether the analysis was of the 'hit-fragment' (which I assumed to be the whole genomic DNA sequence in any one plasmid) or whether the sequence had been edited so only the 3'UTR was included in the analysis. The authors should make it crystal clear whether the sequences they use in the analyses the mRNA maturation sequences or whether they are entirely contained within the 3'UTR.

3. Line 235: a quick explanation of how translational efficiency was measured is necessary

4. In the many available RNAseq datasets how many of the mRNAs >5-fold up regulated in bloodstream forms compared to insect forms contain the U-rich motif (an analysis not an experiment)

5. In the discussion the authors could consider whether it is likely that there is a single mechanism for up regulated mRNAs in bloodstream forms.

Minor points on the manuscript

Line 42: reference for 30% of genome being UTRs

Line 110: what fold increase or decrease was used

Line 125 increased proportion of negative elements from rRNA loci - absence of mRNA maturation signals

Line 147: Is 16/25 'remarkable' given the coverage of the library?

Figure 2B legend: Please specify what the control is. I assume it's the reads from the plasmid library?

Figure 2C: It would be helpful to know whether the fragments identified include the sequences for 3' end maturation.

Line 306: Reference 31 needs to be considered in the context of PMID: 26946399 which provides evidence that the exosomes acts primarily in the nucleus

Line 354: ES promoter here, EP promoter in Figure 1A

Reviewer #2 (Remarks to the Author):

Given that most promoters are believed to be constitutively active in African trypanosomes, most gene regulation takes place post-transcriptionally. However, very little is known about which motifs in the 3' untranslated regions may contribute to gene regulation in this eukaryotic parasite. They conducted a genome-wide screen (UTR-seq) to identify motifs that might influence gene expression. Using machine learning, they discovered that the co/existence of A-rich poly-purine and U-rich poly-pyrimidine in the 3'UTR correlates with higher expression in the bloodstream form of the parasite. This suggests a regulatory mechanism involving the masking and unmasking of poly-purine tracts.

This work offers valuable insights into regulatory motifs and their role in gene regulation in parasites. The methodology is straightforward and effective. However, we recommend enhancements in two areas:

1. Experimental validation is needed to confirm the role of specific motifs in mRNA and/or translation efficiency. Similarly, the proposed model suggesting interaction between poly-pyrimidine and poly-purine could be tested by using reporter genes with the relevant UTRs in bloodstream versus procyclic form trypanosomes.

2. The paper could benefit from clearer explanations and additional information to aid comprehension and assessment of the computational analysis and results:

- 2.1. The cloning strategy requires better explanation. Does the construct have a default a polyadenylation signal downstream of the reporter gene? Since Trypanosoma mRNAs undergo coordinated capping and polyadenylation processes, how is the 3' end of the reporter mRNA processed if no fragment is cloned in or if the cloned fragment does not contain any polyadenylation/trans-splicing recognition sites?

- 2.2. Authors should explain how fold change for positive and negative selection was calculated. Was any statistical approach used or it is simply fold change of normalised counts per fragment over control? Which dataset is the control: plasmid library sequencing, or Trypanosoma library sequencing? Could the authors explain rational?

- 2.3. Please provide quality measures that could help evaluate method robustness, such as the number of reads aligned to the genome in each sample, the number of reads recovered after filtering steps, the total number of identified peaks in reverse/native conformation, the fraction of those peaks deemed significant based on the authors' chosen thresholds, and the number of excluded peaks. Also, the genome browser visualizations do not clearly indicate whether fragments coverage or read coverage is being depicted. It would be more appropriate to visualize fragments, although the figure legend indicates read coverage. Finally, the authors do not explain how library complexity was estimated. It would be useful to know if the Trypanosoma library before induction of the reporter was evaluated/sequenced. While not explicitly stated in the text, we assume that libraries after PCR amplification were not fragmented to preserve the original index sequences. Please clarify.

- 2.4. The number of repeats (n) is not indicated, and it is unclear how different time points were treated in the analysis.

- 2.5. While the use of example representations is useful, it requires additional global data visualizations. For example, correlations between gene expression and fold enrichment obtained in 3'-UTR seq, to underscore the significance of the results overall. Another example in figures 3D and 3E, it is not clear why those four examples of paralogs were selected. If more paralog pairs were recovered by the screen, it would be appropriate to include the analysis of all possible/annotated paralog pairs in a supplementary file.

2.6. Statements, such as "adenine rich regions can be seen in each exemplar 3'-UTR while not seen in 3'-UTRs of genes with lower mRNA abundance," require statistical evaluation.

2.7. In Figure 1B, it's not immediately clear if the reporters used are aldolase and cox subunit. We suggest labelling the plots.

2.8. Line 257's reference to "expression multiple arms of MPRA" is confusing, as there were only two arms.

2.9 Because red-green colour blindness occurs frequently in the population, it would be best to avoid using red and green as contrasting colours in the figures.

Reviewer #3 (Remarks to the Author):

Trenaman et al set out to better understand regulatory non-coding sequences focusing on 3'UTRs found in the African trypanosome. The manuscript presents a well-designed assay for genome scale assessment of sequences that interrogate positive and negative impacts on gene expression in the blood stream stage of the parasite. The authors identify hundreds of such regulatory elements within UTRs and integrate these "hits" with other genomic assays in trypanosomes. Assessment of nucleotide composition of regulatory elements reveals both length and sequence composition bias. The findings are important for understanding regulatory features of the parasite.

Major comments:

1. The design, scale, and read out of the approach should be discussed in more detail within the text. It would be beneficial to the reader if aspects of FigS1 were brought into the main Fig1. The authors should consider including a schematic how the inserted sequences in the reporter were selectively amplified for nextgen sequencing. If possible, the authors should comment on contamination of non-reporter genomic DNA in the sequenced pools and if/how those reads were considered. The use of the indexed reads needs to be expanded upon in the methods sections. Can the authors comments on the statistics used to determine "hits"?

2. While of great value, the manuscript is based entirely on a single experiment. Given that the reporter is based on genomic DNA the reader may be left wondering what portion of a UTR is essential for regulatory control. To ensure the robustness of the assay the authors should consider validating a small subset of elements that were among the top hits.

3. Along similar lines as point 3, can the authors please expand on how genomic coverage from the reporter that spans multiple regions (example 2e, 3c) where dealt with? What proportion of transcript region coverage was used to assign to UTRs vs CDS? That is for fragments that span CDS, UTR, and intergenic where these still assigned to UTR3?

4. Can the authors comment how the inserted regions may impact the processing of the produced reporter transcripts? Is there any concern that polyadenylation may be impacted? Can this be assessed based on the RNAseq?

5. The dual positive and negative selection system is elegant but primarily highlighted in Fig.2. Can the authors comment on overlap between the two screens? Can the authors comments on how one should interpret overlapping genes?

6. Similarly, more details need to be provided with respect to motif analysis of hits as it pertains to the region inserted. Was nucleotide composition of the entire region considered or just the annotated

UTR? In terms of drawing correlations between impact of region in the screen and nucleotide composition this may be important.

7. Can the authors clarify if only indexed reads were ultimately used for determination of enrichment?

8. As a control experiment, one might expect that enriched UTRs would display their impact only when inserted in the sense orientation. It would be helpful to determine what proportion of "hits" in the sense direction had no impact or some impact in the antisense direction. The circos plot is challenging to interpret and summary with stats may be useful.

9. The authors propose RNA structure between As and Us near the end of the manuscript, the manuscript would be strengthened if in silico RNA folding was used to measure the impact of RNA structure on the correlations observed.

10. Can the authors comment on the conservation of "hits" across related organisms? Presumably genome sequences can be used followed by alignments to further support their findings and add an aspect of functional conservation.

Minor Comments:

The methods section with respect to statistics and quantification should be expanded. As noted above a more robust end to end description of library assembly, screening, and analysis should be provided.

Some figure axis labels are very challenging to read because they are small.

While not explicitly required, the manuscript is an analysis of single experiment. The authors could consider additional functional studies to support their claims.

Post-transcriptional reprogramming by thousands of mRNA untranslated regions in trypanosomes

Responses to Reviewers:

Reviewer 1:

In trypanosomes, transcription of protein coding genes is polycistronic with co-transcriptional processing to produce monocistronic mRNAs. There is no obvious wide scale arrangement of genes into functional operons and little or no evidence for regulation of individual genes at the transcriptional level. This places the emphasis on post-transcriptional regulation of mRNA half life and therefore abundance.

The identification of trans- and cis-acting factors that modulate mRNA half life has produced a small number of examples but has not been overly successful.

This manuscript describes a whole genome approach to identifying positive and negative cis-acting elements in bloodstream from *Trypanosoma brucei*. The experimental approach was to clone fragments of genomic DNA downstream of a BSD-TK open reading frame allowing selection with blasticidin for up regulation and with glanciclovir for down regulation. This allowed the identification of fragments of genomic DNA that either increased or decreased the expression of a reporter.

The first figure outlines the screen and shows the outcome as a plot of positive and negative enrichments relative to the library content are shown. Very few positive regulators came from ORFs relative to inter-ORFs and this is illustrated by the distribution of fragments in three dynein genes.

Figure 2 shows: the distribution of 5'UTR, ORF and 3'UTR lengths, the distribution of enrichments for the fragments identified as increasing or decreasing reporter expression highlighting fragments from genes known to be differentially regulated in bloodstream form trypanosomes, five exemplars and the length distribution of 3'UTRs.

Figure 3 shows the base composition bias in different regulatory fragments, the purine rich motif identified as enriched in fragments that increase expression, a set of exemplars showing the location of 'positive and negative regulatory 3'UTR fragments' (Is this the same as the genomic DNA 'hit' or have the 3'UTR sequences been extracted?). Paralogs with different regulation are then analysed and the frequency of the polypurine motif show to correlated with increased expression.

Figure 4 shows the contribution of various features of the mRNA to translational efficiency and show the the 5' and 3'UTRs can be used to give a good prediction of translational efficiency.

Figure 5 considers sequences in the 3'UTR of mRNAs that are unregulated in bloodstream relative to insect stage trypanosomes and shows that a U-rich motif is over-represented in mRNAs upregulated in bloodstream forms and shows 4 exemplars.

As a brief summary, the findings were:

1. High translational efficiency in BSFs was associated with dosage of adenine rich poly purine tracts
2. 3'UTRs associated with upregulated expression in BSFs also enriched in U-rich polypyrimidine tracts

Overall this is elegant work and no further experiments are required. However the authors should consider the following: in interpreting their data.

We thank the reviewer for their comments and excellent suggestions. Regarding their question above, “Is this the same as the genomic DNA ‘hit’ or have the 3’UTR sequences been extracted?”. We apologise that this was only described in the Methods section previously, and now state, in the Results section, “we trimmed hit fragments and retained only those regions that overlapped with reporter-adjacent 3’-UTRs for analysis”.

Comments

1.1: The library screen selected for genomic fragments that resulted in successful maturation (polyadenylation) of the reporter mRNA and any cis-elements that selectively increased or decreased expression. The sequences required for polyadenylation of mRNAs in trypanosomes are complex as it is a two stage process, first the downstream mRNA is trans-spliced and then cleavage and polyadenylation occurs around 100 to 200 nucleotides upstream of the trans-splice site and location does not appear to depend on any sequence motif. Thus, for any fragment to confer expression on the reporter it has to extend well beyond the polyA site of the mRNA. This complex requirement should produce a strand bias. It has previously been shown that the efficiency of trans-splicing, and therefore polyadenylation of the reporter, is sequence dependent (PMID: 16227607) so the expression of the reporter will be affected by the efficiency of mRNA maturation and any 3’UTR elements.

R1.1: All important points that we now deal with in more detail. We’ve added a new paragraph to the Introduction on these points, beginning “In trypanosomatids, long polycistronic transcription units...”. In relation to strand bias - see the new Figure panel 2c and the associated text, beginning “We also expected relatively few mRNA processing...”, and ending “...we observed strong strand-bias following positive ($\chi^2 p = 2.4^{-113}$) or negative ($\chi^2 p = 2.7^{-21}$) selection (Fig. 2c)”. It is possible to produce synthetic or cryptic splice sites, however – see the new text beginning “Notably, in addition to densely packed poly-purine...”. We now cite PMID: 16227607, and include new analysis of downstream splice sites - see new ED Fig. 3 and the associated text beginning, “Since *trans*-splicing-associated sequences...”. At the end of this new paragraph, we state “We conclude that differences in native splice-sites, included downstream of the majority of our hit fragments, had little impact on expression of the reporter in our MPRA”. Our analysis shows that an AC[AG] splice site, which was linked to a 20-fold decrease in *trans* splicing in PMID: 16227607, is actually associated with relatively abundant native transcripts (see ED Fig. 3b). We also note that PMID: 16227607 reports impacts of *trans* splicing on a downstream reporter rather than any impact on an upstream gene.

1.2: In several places I found it hard to distinguish in the text whether the analysis was of the ‘hit-fragment’ (which I assumed to be the whole genomic DNA sequence in any one plasmid) or whether the sequence had been edited so only the 3’UTR was included in the analysis. The authors should make it crystal clear whether the sequences they use in the analyses the mRNA maturation sequences or whether they are entirely contained within the 3’UTR.

R1.2: We apologise that this important detail was only described in the Methods section previously, and now state, in the Results section, just prior to presenting Fig. 4a, “we trimmed hit fragments and retained only those regions that overlapped with reporter-adjacent 3’-UTRs for analysis”.

1.3: Line 235: a quick explanation of how translational efficiency was measured is necessary

R1.3: We’ve added “translation efficiency is calculated by dividing ribosome footprint read-counts by mRNA read-counts for each CDS” here.

1.4: In the many available RNAseq datasets how many of the mRNAs >5-fold up regulated in bloodstream forms compared to insect forms contain the U-rich motif (an analysis not an experiment)

R1.4: The numbers for RNA pol-II transcribed mRNAs associated with 5-fold upregulation in bloodstream forms are: 63 [Jensen *et al.*, 2014; PMID: 25331479], 62 [Hutchinson *et al.*, 2016; PMID: 27756224] and 71 [Naguleswaran *et al.*, 2018; PMID: 29606092]. Numbers associated

with at least one of the current U-rich motifs in the 3'-UTR are 48 (76%), 50 (81%) and 59 (83%), respectively. The relationship between bloodstream upregulated transcripts and U-rich motif density was previously illustrated in Fig. 5b and is now shown in more detail in Fig. 7b-d. We find “significantly higher densities of both the A-rich poly-purine motif shown in Fig. 4b, and the U-rich poly-pyrimidine motif shown in Fig. 7a, in 3'-UTRs associated with bloodstream-form up-translated mRNAs”. We've also now added further analysis for insect-stage upregulated genes, which shows that “Poly-pyrimidine tracts were also significantly enriched in 3'-UTRs associated with more abundant transcripts in insect-stage cells”.

1.5: In the discussion the authors could consider whether it is likely that there is a single mechanism for up regulated mRNAs in bloodstream forms.

R1.5: We feel that the paragraph beginning “mRNA-binding proteins are undoubtedly involved...” deals with this issue. Proteins involved in increasing gene expression in bloodstream forms, ZC3H11, DHH1 and CFB2, as well as others involved in responding to environmental cues, PuREBP1-2 and RBP5, are all mentioned here.

Minor points on the manuscript

1.6: Line 42: reference for 30% of genome being UTRs

R1.6: We've now noted “based on our updated annotation”.

1.7: Line 110: what fold increase or decrease was used

R1.7: This is stated in the Figure legend; “Scale is \log_2 -fold-change relative to plasmid control, with values clipped when >4 ”.

1.8: Line 125 increased proportion of negative elements from rRNA loci - absence of mRNA maturation signals

R1.8: Good point – we've now stated “a lack of mRNA processing signals at *rDNA* loci”.

1.9: Line 147: Is 16/25 ‘remarkable’ given the coverage of the library?

R1.9: We've removed this subjective statement since the *p* values are sufficient here.

1.10: Figure 2B legend: Please specify what the control is. I assume it's the reads from the plasmid library?

R1.10: We revised our statistical analysis such that this panel (now Fig. 3b) shows “indexed read fold-change between the positive and negative selection screens” (also see R2.2.2 below).

1.11: Figure 2C: It would be helpful to know whether the fragments identified include the sequences for 3' end maturation.

R1.11: The majority of hit-fragments do include these sequences and we now state “we found that 79% of blasticidin-selected hits and 59% of ganciclovir-selected hits included the native downstream splice site (Extended Data File 1)”. Also see R1.1 above.

1.12: Line 306: Reference 31 needs to be considered in the context of PMID: 26946399 which provides evidence that the exosomes acts primarily in the nucleus

R1.12: We now state ‘nuclear exosome’ and cite Kramer *et al.*, 2016.

1.13: Line 354: ES promoter here, EP promoter in Figure 1A

R1.13: Thank-you for spotting this error. The label in Fig. 1a has been corrected.

Reviewer 2:

Given that most promoters are believed to be constitutively active in African trypanosomes, most gene regulation takes place post-transcriptionally. However, very little is known about which motifs in the 3' untranslated regions may contribute to gene regulation in this eukaryotic

parasite. They conducted a genome-wide screen (UTR-seq) to identify motifs that might influence gene expression. Using machine learning, they discovered that the co/existence of A-rich poly-purine and U-rich poly-pyrimidine in the 3'UTR correlates with higher expression in the bloodstream form of the parasite. This suggests a regulatory mechanism involving the masking and unmasking of poly-purine tracts.

This work offers valuable insights into regulatory motifs and their role in gene regulation in parasites. The methodology is straightforward and effective. However, we recommend enhancements in two areas:

We thank these reviewers for their positive comments. We would like to clarify here though that our machine learning based predictions indicated “robust correspondence between pPuTs and positive control” rather than addressing developmental controls.

2.1: Experimental validation is needed to confirm the role of specific motifs in mRNA and/or translation efficiency. Similarly, the proposed model suggesting interaction between poly-pyrimidine and poly-purine could be tested by using reporter genes with the relevant UTRs in bloodstream versus procyclic form trypanosomes.

R2.1: Several UTRs and UTR fragments have previously been tested in reporter assays by others, and our results are consistent with prior findings in this regard – see Fig. 3b-e, and the two paragraphs beginning “To further explore the quality and coverage of our dataset...”; and Fig. 7e-f, and the paragraph beginning “The relative density of U-rich and A-rich motifs...”. We accept that further work will be required to test specific hypotheses, but do feel that these studies are beyond the scope of the current manuscript.

2.2: The paper could benefit from clearer explanations and additional information to aid comprehension and assessment of the computational analysis and results:

2.2.1: The cloning strategy requires better explanation. Does the construct have a default polyadenylation signal downstream of the reporter gene? Since Trypanosoma mRNAs undergo coordinated capping and polyadenylation processes, how is the 3' end of the reporter mRNA processed if no fragment is cloned in or if the cloned fragment does not contain any polyadenylation/trans-splicing recognition sites?

R2.2.1: We apologise for omitting this important detail. We now state, “Given the absence of downstream mRNA processing signals in our reporter construct...” and have moved a panel describing the cloning strategy from the extended data to Fig. 1c. Also see R1.1 and R1.11 above.

2.2.2: Authors should explain how fold change for positive and negative selection was calculated. Was any statistical approach used or it is simply fold change of normalised counts per fragment over control? Which dataset is the control: plasmid library sequencing, or Trypanosoma library sequencing? Could the authors explain rationale?

R2.2.2: We have now added manual curation of fragments and added statistical analysis rather than simply using fold-change to define ‘hits’. See the text beginning “All inter-CDS peaks recovered from the MPRA were manually curated...” and “We also employed the Wilcoxon rank-sum test...” in the Results; and the text beginning “We then manually curated the boundaries ...” and the “Statistical analysis” section in the Methods.

2.2.3: Please provide quality measures that could help evaluate method robustness, such as the number of reads aligned to the genome in each sample, the number of reads recovered after filtering steps, the total number of identified peaks in reverse/native conformation, the fraction of those peaks deemed significant based on the authors' chosen thresholds, and the number of excluded peaks. Also, the genome browser visualizations do not clearly indicate whether fragments coverage or read coverage is being depicted. It would be more appropriate to visualize fragments, although the figure legend indicates read coverage. Finally, the authors do not explain how library complexity was estimated. It would be useful to know if the

Trypanosoma library before induction of the reporter was evaluated/sequenced. While not explicitly stated in the text, we assume that libraries after PCR amplification were not fragmented to preserve the original index sequences. Please clarify.

R2.2.3: We've included more detail here, including "reads were mapped to the *T. brucei* genome; 100.3 million paired reads for the blasticidin-selected samples (4.8% with an index sequence), 176.2 million for the ganciclovir-selected samples (4.8% with an index sequence), and 50.7 million for the plasmid library (6.5% with an index sequence)", "Positive selection with blasticidin yielded 80 enriched CDS peaks and 1,827 enriched 'inter-CDS' peaks, while negative selection with ganciclovir yielded 918 enriched CDS peaks and 1,915 enriched 'inter-CDS' peaks", and "This assessment revealed significant enrichment (FDR <0.1) for 1,112 positive regulatory hit fragments from 1,070 3'-UTRs, and 807 negative regulatory hit fragments from 801 3'-UTRs". For the genome visualisations, we show sense paired, indexed reads to "highlight the boundaries of hit fragments..."; see new detail in the Fig. 2b legend. For the library complexity estimation, we've added a panel to ED Fig. 1c to clarify. We did not evaluate the *T. brucei* library prior to induction. Libraries were fragmented, and we've now added further detail; see the text beginning, "PCR amplicons were subjected to tagmentation to produce 300-350 bp fragments...".

2.2.4: The number of repeats (n) is not indicated, and it is unclear how different time points were treated in the analysis.

R2.2.4: Our new statistical analysis involved comparing three sample pairs, "and indexed reads from opposite ends of each fragment were considered separately"; see R2.2.2 above and the new "Statistical analysis" section in the Methods.

2.2.5: While the use of example representations is useful, it requires additional global data visualizations. For example, correlations between gene expression and fold enrichment obtained in 3'-UTR seq, to underscore the significance of the results overall. Another example in figures 3D and 3E, it is not clear why those four examples of paralogs were selected. If more paralog pairs were recovered by the screen, it would be appropriate to include the analysis of all possible/annotated paralog pairs in a supplementary file.

R2.2.5: An excellent suggestion. We've added an analysis of fold enrichment relative to measures of translation efficiency, mRNA abundance or mRNA turnover – see Fig. 4c, Extended Data Fig. 4a-b, and the paragraph beginning "Selection in our MPRA was contingent upon..."; "We found that hits in the MPRA were more strongly correlated with differences in translation efficiency (Fig. 4c, $p = 1.2^{-68}$)...". We've now carried out a systematic search for tandem 'paralogous pair hits', which revealed nine with differential pPuT abundance. See the revised section entitled "Poly-purine tracts are enriched in the UTRs of highly expressed paralogs", and revised Fig. 5. We now state that, "poly-purine tract density and 3'-UTR length are predictive of differential translation for all of these paralog-pairs".

2.2.6: Statements, such as "adenine rich regions can be seen in each exemplar 3'-UTR while not seen in 3'-UTRs of genes with lower mRNA abundance," require statistical evaluation.

R2.2.6: Similar to the point above. We've added an analysis of A-rich motifs relative to measures of translation efficiency, mRNA abundance or mRNA turnover – see Fig. 4d, Extended Data Fig. 4c-d, and the paragraph beginning "A similar analysis, but this time...". The analysis "revealed a clear correlation with translation efficiency (Fig. 4d, Pearson = 0.36)".

2.2.7: In Figure 1B, it's not immediately clear if the reporters used are aldolase and cox subunit. We suggest labelling the plots.

R2.2.7: We've added labels as suggested.

2.2.8: Line 257's reference to "expression multiple arms of MPRA" is confusing, as there were only two arms.

R2.2.8: This referred to the positive, negative and 'synthetic' arms of the screen but "multiple arms of" has now been removed.

2.2.9: Because red-green colour blindness occurs frequently in the population, it would be best to avoid using red and green as contrasting colours in the figures.

R2.2.9: We thank the reviewer(s) for this suggestion and have adjusted the colours to make our Figures more accessible.

Reviewer 3:

Trenaman et al set out to better understand regulatory non-coding sequences focusing on 3'UTRs found in the African trypanosome. The manuscript presents a well-designed assay for genome scale assessment of sequences that interrogate positive and negative impacts on gene expression in the blood stream stage of the parasite. The authors identify hundreds of such regulatory elements within UTRs and integrate these “hits” with other genomic assays in trypanosomes. Assessment of nucleotide composition of regulatory elements reveals both length and sequence composition bias. The findings are important for understanding regulatory features of the parasite.

We thank the reviewer for their positive comments and suggestions.

Major comments:

3.1: The design, scale, and read out of the approach should be discussed in more detail within the text. It would be beneficial to the reader if aspects of FigS1 were brought into the main Fig1. The authors should consider including a schematic how the inserted sequences in the reporter were selectively amplified for nextgen sequencing. If possible, the authors should comment on contamination of non-reporter genomic DNA in the sequenced pools and if/how those reads were considered. The use of the indexed reads needs to be expanded upon in the methods sections. Can the authors comments on the statistics used to determine “hits”?

R3.1: We've included more detail here (see R2.2.3), explained “how library complexity and genome coverage were calculated” (ED Fig. 1c), and have moved a panel describing the cloning strategy from the extended data to Fig. 1c. We hope that this panel, alongside the text stating “Genomic DNA was extracted from each selected sample, and DNA libraries were generated by amplifying the fragments cloned immediately downstream of the reporter, including the flanking index sequences (Fig. 1b...” is sufficient here. Regarding indexed reads and statistics – see R2.2.2 and R2.2.4 above.

3.2: While of great value, the manuscript is based entirely on a single experiment. Given that the reporter is based on genomic DNA the reader may be left wondering what portion of a UTR is essential for regulatory control. To ensure the robustness of the assay the authors should consider validating a small subset of elements that were among the top hits.

R3.2: See R2.1 above - Several UTRs and UTR fragments have previously been tested in reporter assays by others, and our results are consistent with prior findings in this regard – see Fig. 3b-e, and the two paragraphs beginning “To further explore the quality and coverage of our dataset...”; and Fig. 7e-f, and the paragraph beginning “The relative density of U-rich and A-rich motifs...”. We accept that further work will be required to test specific hypotheses, but do feel that these studies are beyond the scope of the current manuscript. We also feel that our massive parallel reporter assay, combined with machine learning based predictions (Fig. 6), and analysis of synthetic 3'-UTRs (Fig. 8), provides robust support for our main conclusions, and also that the new analysis we now include substantially strengthens the case.

3.3: Along similar lines as point 3, can the authors please expand on how genomic coverage from the reporter that spans multiple regions (example 2e, 3c) where dealt with? What proportion of transcript region coverage was used to assign to UTRs vs CDS? That is for fragments that span CDS, UTR, and intergenic where these still assigned to UTR3?

R3.3: This is described in the Methods section under ‘UTR library sequencing and analysis’ (previously lines 419-424) – see text beginning “we assigned and trimmed blasticidin and ganciclovir fragments...”, and ending “...we selected the 3'-UTR overlapping segment closest

to the *BSD-TK* CDS in the reporter construct". We've also now stated "we trimmed significantly enriched hit fragments and retained only those reporter-adjacent regions that overlapped with 3'-UTRs for analysis (Extended Data File 1)" in the Results section.

3.4: Can the authors comment how the inserted regions may impact the processing of the produced reporter transcripts? Is there any concern that polyadenylation may be impacted? Can this be assessed based on the RNAseq?

R3.4: See R1.1 and R1.11 above: The majority of hit-fragments include downstream processing signals and we now state "we found that 79% of blasticidin-selected hits and 59% of ganciclovir-selected hits included the native downstream splice site (Extended Data File 1)".

3.5: The dual positive and negative selection system is elegant but primarily highlighted in Fig.2. Can the authors comment on overlap between the two screens? Can the authors comments on how one should interpret overlapping genes?

R3.5: Among significantly enriched hit fragments, we find 27 3'-UTRs with fragments from both the positive and negative screens (1.4% of 3'-UTRs with hit-fragments). Developmentally regulated genes (including 10.8490, THT2 in Fig. 3e) are enriched in this set ($\chi^2 p = 0.036$). One possible explanation is that 3'-UTRs associated with developmentally regulated genes are often longer than average. Another is that UTRs associated with bloodstream form upregulated genes are enriched in both A-rich and U-rich regions (see Fig. 7). Some of these hit fragments may equally represent background noise in the assay. Since all of these features may contribute, we'd rather not speculate on this point in the paper.

3.6: Similarly, more details need to be provided with respect to motif analysis of hits as it pertains to the region inserted. Was nucleotide composition of the entire region considered or just the annotated UTR? In terms of drawing correlations between impact of region in the screen and nucleotide composition this may be important.

R3.6: See R3.3 above. We now state "we trimmed hit fragments and retained only those regions that overlapped with reporter-adjacent 3'-UTRs for analysis (Extended Data File 1)", immediately before describing the nucleotide composition and motif analysis shown in Fig. 4a-b. Subsequent analysis focuses on 'A-rich motifs in 3'-UTR' (Fig. 4d) or 'A-rich motif density in 3'-UTR' (Fig. 4e, Fig. 5b, Fig. 7b-d).

3.7: Can the authors clarify if only indexed reads were ultimately used for determination of enrichment?

R3.7: These details are described in the Methods section under 'UTR library sequencing and analysis' (previously lines 432-435) – see text beginning "Fold-changes in indexed reads...", and are now also detailed in the Results, "We derived counts for indexed paired reads and ran pairwise comparisons for day-4, day-6 and day-8 positive and negative selected samples to calculate \log_2 fold changes.". Also see R2.2.2 and R2.2.4 above.

3.8: As a control experiment, one might expect that enriched UTRs would display their impact only when inserted in the sense orientation. It would be helpful to determine what proportion of "hits" in the sense direction had no impact or some impact in the antisense direction. The circos plot is challenging to interpret and summary with stats may be useful.

R3.8: See R1.1; "we observed strong strand-bias following positive ($\chi^2 p = 2.4^{-113}$) or negative ($\chi^2 p = 2.7^{-21}$) selection (Fig. 2c)". Also see R2.2.3; we've added specific details here - "Positive selection with blasticidin yielded 80 enriched CDS peaks and 1,827 enriched 'inter-CDS' peaks, while negative selection with ganciclovir yielded 918 enriched CDS peaks and 1,915 enriched 'inter-CDS' peaks".

3.9: The authors propose RNA structure between As and Us near the end of the manuscript, the manuscript would be strengthened if in silico RNA folding was used to measure the impact of RNA structure on the correlations observed.

R3.9: Experimental analysis and computational prediction of RNA secondary structure are active, but challenging research areas. Unfortunately, RNA folding predictions are often inaccurate, particularly for RNA of >200 nucleotides, and typically fail to reflect folding under biological conditions (reviewed by Zhang *et al.*, 2022 - PMID: 36203019). We ran some preliminary analysis using the '*RNAfold*' tool but this was not fruitful. We've added text to the Discussion stating "Modulation of RNA folding, structure, and function may also be impacted by mRNA-binding proteins, or long non-coding RNAs".

3.10: Can the authors comment on the conservation of "hits" across related organisms? Presumably genome sequences can be used followed by alignments to further support their findings and add an aspect of functional conservation.

R3.10: Analysis of related trypanosomatids is challenging in the absence of accurate 3'-UTR annotations for these genomes. Indeed, we "revised or added 3'-UTR annotations for 4,703 genes" as part of the current study; see the Methods section, entitled "Annotation of 3'-UTRs".

Minor Comments:

3.11: The methods section with respect to statistics and quantification should be expanded. As noted above a more robust end to end description of library assembly, screening, and analysis should be provided.

R3.11: See R2.2.3, R2.2.4 and R3.1 above.

3.12: Some figure axis labels are very challenging to read because they are small.

R3.12: We've adjusted the axis labels in Fig. 2b (now Fig. 3b) and for all gene maps in Fig's 3-5 and 7.

3.13: While not explicitly required, the manuscript is an analysis of single experiment. The authors could consider additional functional studies to support their claims.

R3.13: See R3.2 above.

REVIEWER COMMENTS

Reviewer #1 (Remarks to the Author):

The authors have answered all the comments made in my first review.

Reviewer #2 (Remarks to the Author):

The authors identified thousands of 3'-UTRs with the potential to improve gene expression. This work undoubtedly offers a valuable resource for the Trypanosoma research community and will likely be explored further by others.

Although the authors have not experimental validated novel regulatory UTRs, they show convincing evidence that their results are consistent with previously published findings.

A drawback of the method is that since the readout depends on protein product, the impact of UTR on mRNA translation or/and stability cannot be distinguished.

The authors made a substantial effort to improve the clarity of their manuscript in response to reviewer comments. While acknowledging their work, two weaknesses remain:

1. Evidence for U-rich poly-pyrimidine tracts function: The authors provide solid evidence that A-rich poly-purine tracts correlate positively with enhanced translation. Similarly, one can agree that U-rich tracts associate with genes that are highly expressed in blood-stream form. However, the hypothesis that poly-pyrimidine tracts can regulate transcript translatability negatively through their coordinated action with purine stretches is speculative and, in my opinion, has weak support in the data provided. For instance, in Figure 7c (left-hand panels), both U-rich and A-rich motifs appear to enhance RNA translatability in both life stages, whereas examples (only 2!) selected for the right panel might suggest that U-stretches are absent from RNAs highly expressed in pro-cyclic forms.

As such I do not fully agree with the statements from lines 372-373 or 390-392

Since the authors declined to perform additional experiments to validate their hypothesis, I believe it should either be removed from the text or moved to the discussion section, where it should be clearly stated as speculation.

2. Lack of Detail in the Proposed Model: The model proposed for A-rich poly-purine stretches lacks detail. Could the authors elaborate more on potential mechanism? Are there examples from other systems where A-rich sequences enhance/regulate translation? Authors suggest that the translation machinery can bind to these sequences. What exactly do they mean by this? Could poly(A)-binding protein bind to such sequences?

Minor Comments on the revised manuscript:

- Line 339: "Longer 3'-UTRs with a high dosage and density of both poly-purine and poly-pyrimidine tracts were primarily associated with bloodstream-form up-regulated transcripts (Fig. 7c, left-hand panels)."

I am unsure about this conclusion since the number of transcripts between the two groups is very different. Could the authors justify this differently? Are the observed distributions statistically different?

- Variable Fold Change Thresholds: In the chapter "Poly-purine and -pyrimidine rich 3'-UTRs in bloodstream up-regulated mRNAs," the authors use different fold change thresholds depending on the

analysis/visualizations they are performing. Could the threshold be consistent, and if not, could the authors justify their choices for different analyses?

- Line 124: Could the authors explain why only a few percent of reads had index sequences?

- Line 262: The authors discuss Pearson correlation analysis. In my opinion, correlations with R-squared values below 0.1 are very low and can be considered negligible. I would refer to these as "weak" rather than "modest."

Reviewer #3 (Remarks to the Author):

The authors have made many changes to improve the manuscript. Especially those dealing with providing more detail on methodology and analysis.

However, the entire manuscript hinges on a high-throughput experiment and it would still be of benefit to validate novel hits to improve the rigor of this study. To their credit the authors do comment on the fact that known regulatory UTRs are also found in their screen. But as might be expected the overlap is not perfect not always in the expected direction. While using previously known regulatory elements to calibrate their study is very useful and provides valuable insights, these UTRs do not represent novel validated targets among the thousands that were tested. At the very least a discussion on lack of validation of novel hits should be included and highlighted as a limitation of the approach and results and not only discussed as a future direction. In a more ideal scenario the authors would validate top novel hits.

Reviewer #4 (Remarks to the Author):

Reviewer #4 (Remarks on code availability):

While I did not run the code, the code is available and looks credible.

Post-transcriptional reprogramming by thousands of mRNA untranslated regions in trypanosomes
Nature Communications.

Reviewer #1:

The authors have answered all the comments made in my first review.

Reviewer #2:

The authors identified thousands of 3'-UTRs with the potential to improve gene expression. This work undoubtedly offers a valuable resource for the Trypanosoma research community and will likely be explored further by others.

Although the authors have not experimentally validated novel regulatory UTRs, they show convincing evidence that their results are consistent with previously published findings.

A drawback of the method is that since the readout depends on protein product, the impact of UTR on mRNA translation or/and stability cannot be distinguished.

The authors made a substantial effort to improve the clarity of their manuscript in response to reviewer comments. While acknowledging their work, two weaknesses remain:

1. Evidence for U-rich poly-pyrimidine tracts function: The authors provide solid evidence that A-rich poly-purine tracts correlate positively with enhanced translation. Similarly, one can agree that U-rich tracts associate with genes that are highly expressed in bloodstream form. However, the hypothesis that poly-pyrimidine tracts can regulate transcript translatability negatively through their coordinated action with purine stretches is speculative and, in my opinion, has weak support in the data provided. For instance, in Figure 7c (left-hand panels), both U-rich and A-rich motifs appear to enhance RNA translatability in both life stages, whereas examples (only 2!) selected for the right panel might suggest that U-stretches are absent from RNAs highly expressed in procyclic forms. As such I do not fully agree with the statements from lines 372-373 or 390-392. Since the authors declined to perform additional experiments to validate their hypothesis, I believe it should either be removed from the text or moved to the discussion section, where it should be clearly stated as speculation.

For clarification, the two examples of procyclic form up-translated transcripts shown on the right in Fig. 7c and in Fig. 7e, reflect relatively low A-rich motif density (as also seen in Fig. 7b), rather than absence of U-stretches. When describing evidence for poly-pyrimidine tract function, we state “*suggesting* a mechanism for developmental activation through pPuT ‘unmasking’” in the Abstract, “*suggest* developmental modulation of poly-purine tract function by poly-pyrimidine tracts” at the end of the Introduction, and “These findings *suggest* that a high density of poly-pyrimidine tracts reduces translation (and mRNA stability) in bloodstream-form trypanosomes” in the “Poly-purine and -pyrimidine...” Results section. We also state “Although our MPRA was not designed to assess developmental controls, our analysis also suggests that poly-pyrimidine tracts can conditionally modulate the activity of poly-purine tracts” and “We *suggest* that conditional unmasking of poly-purine tracts contributes to increased expression in bloodstream-form cells” in the Discussion. We agree that our hypothesis regarding poly-pyrimidine tracts is speculative, and feel that our use of ‘*suggest(ing)*’ etc serves to reflect this view. We’ve now added text on “Limitations of our study...” in the Discussion (see response to Reviewer 3 below), and reiterate here “that our hypothesis regarding developmental control through (un)masking of poly-purine tracts is more speculative”. We’ve also adjusted the text from lines 372-373, now stating “*suggested* a role for poly-pyrimidine tracts in modulating poly-purine tract function”. We are unsure as to how

to respond in relation to the text from lines 390-392, however, as this text does not refer to poly-pyrimidine tract function.

2. Lack of Detail in the Proposed Model: The model proposed for A-rich poly-purine stretches lacks detail. Could the authors elaborate more on potential mechanism? Are there examples from other systems where A-rich sequences enhance/regulate translation? Authors suggest that the translation machinery can bind to these sequences. What exactly do they mean by this? Could poly(A)-binding protein bind to such sequences?

In the Discussion, “we propose recruitment of translation machinery by positive regulatory A-rich poly-purine tracts”. It is possible that poly(A)-binding proteins bind to such sequences. Recruitment of translation machinery may also be indirect, may involve base-pairing, or even interactions with long non-coding RNAs. We prefer not to speculate on these points, however. Rather, we hope that the next paragraph, beginning “Limitations of our study...”, serves to illustrate the need for further work in this area. Regarding examples from other systems where A-rich sequences enhance/regulate translation, we state, towards the end of the Discussion, “gene expression is primarily controlled at the level of translation in human cells⁶⁰, where purine-rich 5'-UTRs can promote translation factor binding and translation⁶¹”.

Minor Comments on the revised manuscript:

- Line 339: "Longer 3'-UTRs with a high dosage and density of both poly-purine and poly-pyrimidine tracts were primarily associated with bloodstream-form up-regulated transcripts (Fig. 7c, left-hand panels)."

I am unsure about this conclusion since the number of transcripts between the two groups is very different. Could the authors justify this differently? Are the observed distributions statistically different?

We thank the reviewer for raising this point and have adjusted this text to “this cohort of bloodstream-form up-translated transcripts was specifically overrepresented ($\chi^2 p = 1.8 \cdot 10^{-7}$) for longer (>2 kbp) 3'-UTRs with a high density (>20 per kbp) of both poly-purine and poly-pyrimidine tract motifs”. We’ve also added further statistical analyses in the Fig. 7b panels.

- Variable Fold Change Thresholds: In the chapter "Poly-purine and -pyrimidine rich 3'-UTRs in bloodstream up-regulated mRNAs," the authors use different fold change thresholds depending on the analysis/visualizations they are performing. Could the threshold be consistent, and if not, could the authors justify their choices for different analyses?

We used ‘>4-fold’ for Fig. 7b-c to capture what we considered to be a sufficient number (81) of procyclic form up-translated transcripts for analysis; <50 transcripts were >5-fold up-translated in procyclic form cells. Similarly, we used ‘>3-fold’ for the more focussed analysis of “up-translated transcripts in bloodstream-form cells, with 3'-UTRs of >2 kbp and with >20 poly-purine and poly-pyrimidine tract motifs per kbp”; <30 transcripts were >5-fold up-translated in this cohort.

- Line 124: Could the authors explain why only a few percent of reads had index sequences? Fig. 1b indicates the locations of index sequences relative to 1-3 kbp genomic DNA fragments. We also state in the Methods section, “PCR amplicons were subjected to tagmentation to produce 300-350 bp fragments ... The libraries were then pooled and sequenced on a NextSeq 500 (Illumina) platform to obtain 144-228 million 75-b paired-end reads per sample”.

- Line 262: The authors discuss Pearson correlation analysis. In my opinion, correlations with R-squared values below 0.1 are very low and can be considered negligible. I would refer to these as “weak” rather than “modest.”

Adjusted as suggested.

Reviewer #3:

The authors have made many changes to improve the manuscript. Especially those dealing with providing more detail on methodology and analysis.

However, the entire manuscript hinges on a high-throughput experiment and it would still be of benefit to validate novel hits to improve the rigor of this study. To their credit the authors do comment on the fact that known regulatory UTRs are also found in their screen. But as might be expected the overlap is not perfect not always in the expected direction. While using previously known regulatory elements to calibrate their study is very useful and provides valuable insights, these UTRs do not represent novel validated targets among the thousands that were tested. At the very least a discussion on lack of validation of novel hits should be included and highlighted as a limitation of the approach and results and not only discussed as a future direction. In a more ideal scenario the authors would validate top novel hits.

We've added the following paragraph to the Discussion:

“Limitations of our study include a readout from our reporter assay that depended upon protein product, meaning that impacts of a 3'-UTR on mRNA maturation, translation, and turnover were not assessed separately, as well as lack of independent validation of novel hit fragments and 3'-UTRs. Our reporter assay was also conducted using only bloodstream form cells, meaning that our hypothesis regarding developmental control through (un)masking of poly-purine tracts is more speculative. Future studies in this area should address these limitations and may reveal regulatory factors that control expression by interacting with poly-purine and/or poly-pyrimidine tracts, and/or other regulatory sequences, in 5'-UTRs and/or 3'-UTR.”

Reviewer #4:

Remarks on code availability:

While I did not run the code, the code is available and looks credible.